# Exploration-Guided Reward Shaping
# for Reinforcement Learning under Sparse Rewards

**Rati Devidze**[1]
rdevidze@mpi-sws.org

**Parameswaran Kamalaruban**[2]
kparameswaran@turing.ac.uk

**Adish Singla**[1]
adishs@mpi-sws.org

[1]Max Planck Institute for Software Systems (MPI-SWS), Saarbrucken, Germany
[2]The Alan Turing Institute, London, UK

## Abstract

We study the problem of reward shaping to accelerate the training process of a reinforcement learning agent. Existing works have considered a number of different reward shaping formulations; however, they either require external domain knowledge or fail in environments with extremely sparse rewards. In this paper, we propose a novel framework, Exploration-Guided Reward Shaping (EXPLORS), that operates in a fully self-supervised manner and can accelerate an agent's learning even in sparse-reward environments. The key idea of EXPLORS is to learn an intrinsic reward function in combination with exploration-based bonuses to maximize the agent's utility w.r.t. extrinsic rewards. We theoretically showcase the usefulness of our reward shaping framework in a special family of MDPs. Experimental results on several environments with sparse/noisy reward signals demonstrate the effectiveness of EXPLORS.

## 1   Introduction

Training reinforcement learning (RL) agents in environments with extremely sparse or distracting rewards is challenging. Existing works have studied several approaches to design informative rewards that speed up the agent's convergence [1–7]. One well-studied line of work is potential-based reward shaping, where a potential function is specified by an expert or obtained via transfer learning techniques (see [3, 8–17]). Another popular approach is to learn rewards via Inverse-RL using expert demonstrations [18]. Alternatively, one could also consider a manual specification of rewards, e.g., using distance-based metrics [19]. However, these reward design techniques typically rely on high-quality domain knowledge and may fail in practice. In fact, the RL agents can easily exploit poorly designed rewards and get stuck in local optima. This naturally leads to the fundamental question of how to do online reward shaping without relying on expert domain knowledge. More concretely, *can we design informative rewards that will accelerate the agent's training process by leveraging experience gained online during the agent's training lifetime itself?* [20–24]

To tackle this question, recent works [24–26] have explored fully self-supervised learning of parametric intrinsic rewards that can improve the performance of RL agents. In particular, these methods alternate between intrinsic reward parameter learning and the agent's policy optimization w.r.t. the learned reward. For instance, Learning Intrinsic Rewards for Policy Gradient (LIRPG) technique [25] updates the intrinsic reward parameters to maximize the extrinsic rewards received by the policy from the environment. Self-supervised Online Reward Shaping (SORS) technique [26] infers an intrinsic reward using a classification-based reward inference algorithm, TREX [27]. However, these fully self-supervised reward shaping techniques might fail to produce meaningful agent behavior in environments with extremely sparse rewards (called *hard-exploration* domains) as they lack an

36th Conference on Neural Information Processing Systems (NeurIPS 2022).

explicit explorative component. Intuitively, these techniques will not be able to make updates to parameters of their intrinsic reward functions, without receiving a non-zero extrinsic reward signal.

In a parallel line of work, several techniques have been proposed to specifically tackle the challenges of extreme sparsity and exploration. One such line of work is to add more stochasticity in the agent's behavior (e.g., [28–30]); however such techniques typically succeed in tasks with already well-shaped rewards. Another important line of work, relevant to our proposed framework, is bonus-driven exploration techniques for tackling hard-exploration domains – these techniques augment the extrinsic rewards with additional intrinsic bonus signals to encourage extra exploration [31]. A popular category of intrinsic bonuses is count-based bonuses that encourage RL agents to experience infrequently visited states [32–34]. Another category of intrinsic bonuses is providing rewards for improving the agent's knowledge about the environment [35–40]. However, simply relying on these bonus-driven signals can mislead the agent towards sub-optimal or bad behaviors — for instance, in *noisy-distractive* domains such as the "noisy TV" problem [41], unpredictable random or noisy outputs would attract the agent's attention forever.

An important research question that we seek to address is: *How can we design an online intrinsic reward function, without any domain knowledge, that can speed up the agent's learning process even in environments with extremely sparse rewards and noisy distractions?* To this end, we propose a novel framework, Exploration-Guided Reward Shaping (EXPLORS), that learns an intrinsic reward function in combination with exploration-based bonuses to maximize the agent's utility. EXPLORS operates in a fully self-supervised manner, and alternates between reward learning and policy optimization. Our main results and contributions are:

I. We propose a novel reward shaping framework, EXPLORS, that operates in a fully self-supervised manner and can accelerate an agent's learning even in sparse-reward environments. (Section 3.1).

II. We derive intuitive meta-gradients for updating the intrinsic reward component of EXPLORS that enables our framework to be broadly applicable to any RL agent and not only policy-gradient based agents (Sections 3.2 and 3.3).

III. We theoretically showcase the usefulness of our reward shaping framework in accelerating an agent's learning in a special family of chain environments (Section 3.4).

IV. We empirically demonstrate the effectiveness of EXPLORS on several environments with sparse and noisy reward signals (Section 4).[1]

## 2 Problem Setup

In Section 2.1, we present a general framework of online reward shaping technique for RL agents. In Section 2.2, we discuss the limitations of existing reward shaping techniques.

### 2.1 General Framework of Online Reward Shaping

**Preliminaries.** An environment is defined as a Markov Decision Process (MDP) $M :=(\mathcal{S}, \mathcal{A}, T, P_0, \gamma, R)$, where the state and action spaces are denoted by $\mathcal{S}$ and $\mathcal{A}$ respectively. $T : \mathcal{S} \times \mathcal{S} \times \mathcal{A} \rightarrow [0, 1]$ captures the state transition dynamics, i.e., $T(s' \mid s, a)$ denotes the probability of landing in state $s'$ by taking action $a$ from state $s$. $\gamma$ is the discounting factor, and $P_0$ is the initial state distribution. The reward function is given by $R : \mathcal{S} \times \mathcal{A} \rightarrow [-R_{\max}, R_{\max}]$, for some $R_{\max} > 0$. We denote the true underlying extrinsic reward function by $\overline{R}$ and the designed reward function by $\widehat{R}$. We denote a stochastic policy $\pi : \mathcal{S} \rightarrow \Delta(\mathcal{A})$ as a mapping from a state to a probability distribution over actions, and a deterministic policy $\pi : \mathcal{S} \rightarrow \mathcal{A}$ as a mapping from a state to an action. For any trajectory $\xi = \{(s_t, a_t)\}_{t=0,1,...,H}$, we define its cumulative return w.r.t. reward function $R$ as $J(\xi, R) := \sum_{t=0}^{H} \gamma^t \cdot R(s_t, a_t)$. Then, the expected cumulative return (value) of a policy $\pi$ w.r.t. $R$ is defined as $J(\pi, R) := \mathbb{E}[J(\xi, R)|P_0, T, \pi]$, where $s_0 \sim P_0(\cdot)$, $a_t \sim \pi(\cdot|s_t)$, and $s_{t+1} \sim T(\cdot|s_t, a_t)$. The learner seeks to find a policy that has maximum value w.r.t. the extrinsic reward function $\overline{R}$, i.e., $\max_\pi J(\pi, \overline{R})$.

---

[1]Github repo: https://github.com/machine-teaching-group/neurips2022_exploration-guided-reward-shaping.

---
**Algorithm 1** Online Reward Shaping
---
1: **Input:** Extrinsic reward $\overline{R}$, and RL algorithm $L$
2: **Initialization:** $\pi_0, \widehat{R}_0$
3: **for** $k = 1, 2, \ldots, K$ **do**
4:     update policy $\pi_k \leftarrow L(\pi_{k-1}, \widehat{R}_{k-1})$
5:     update reward $\widehat{R}_k$ using $\widehat{R}_{k-1}$ and $\pi_k$
6: **Output:** $\pi_K$
---

**Online reward shaping.** A general framework of online reward shaping for RL agents is given in Algorithm 1. A natural objective here is to design informative rewards $\widehat{R}_k$ at each round $k$ so that the resulting final policy $\pi_K$ performs better (i.e., has high value w.r.t. $\overline{R}$) compared to the corresponding policy obtained via the standard training with $\widehat{R}_k = \overline{R}$. Note that we consider a single lifetime training setting for an RL agent on a single task, i.e., there is no resetting of the policy between rounds.

## 2.2 Existing Techniques and Issues

A popular technique for reward shaping is potential-based reward shaping (PBRS) which guarantees that any optimal policy induced by the designed reward function is also optimal under the extrinsic reward function [3]. However, for PBRS to be effective in accelerating the training process of an RL agent, we need to have access to good potential functions based on expert domain knowledge [42]. The focus of our work is on designing fully self-supervised reward shaping techniques. Below, we provide a discussion of existing techniques that do not require any expert guidance or domain knowledge, and also discuss their limitations.

**Reward shaping based on exploration bonuses.** In the bonus-driven exploration framework [32–34], a count-based intrinsic bonus $B_k(s)$ is given to the agent to encourage exploration. The bonus $B_k(s)$ measures the "novelty" of a state $s$ given the history of all transitions up to round $k$. The authors in [34] extend the classic exploration methods with count-based intrinsic bonuses [43–46] to high-dimensional, continuous state spaces. However, these "exploration-only" reward shaping techniques do not appropriately combine the successful extrinsic reward signals received from the environment. When there are distractive zones in the state space, these methods will keep on exploring the state space even after obtaining extrinsic reward signals.

**Fully self-supervised reward shaping: LIRPG [25].** Learning Intrinsic Rewards for Policy Gradient (LIRPG) technique [25] considers a parametric reward function of the form $\widehat{R}^{\text{LIRPG}}(s, a) = \overline{R}(s, a) + R_\phi(s, a)$, and learns the parameter $\phi$ of the intrinsic reward function $R_\phi$ in a fully self-supervised manner. LIRPG alternates between learning the intrinsic reward parameter $\phi$ and the agent's policy optimization w.r.t. the learned reward $\widehat{R}^{\text{LIRPG}}$. At round $k$, for fixed $\pi_k$, LIRPG updates the parameter $\phi_{k-1}$ to $\phi_k$ by considering the effect such a change would have on the expected cumulative return (w.r.t. $\overline{R}$) of the learner through the change in the policy $\pi_k$, i.e., update $\phi$ using the gradient $\left[ \nabla_\phi J(L(\pi_k, \widehat{R}^{\text{LIRPG}}), \overline{R}) \right]_{\phi_{k-1}}$. In order to develop an update rule for $\phi$, LIRPG considers policy gradient style learning algorithm $L$ with parametric policies $\left\{ \pi_\theta : \theta \in \mathbb{R}^{d_\theta} \right\}$. More concretely, for a parameter $\theta_k$ at round $k$ s.t. $\pi_k := \pi_{\theta_k}$, the learner's policy update depends on $\phi$ as $L(\pi_k, \widehat{R}^{\text{LIRPG}}) := \pi_{\theta(\phi)}$, where $\theta(\phi) = \theta_k + \alpha \cdot \left[ \nabla_\theta J(\pi_\theta, \widehat{R}^{\text{LIRPG}}) \right]_{\theta_k}$. Based on this learner update, the LIRPG update for the intrinsic reward parameters, at round $k$, is based on the following meta-gradients: $\phi_k = \phi_{k-1} + \eta \cdot \left[ \nabla_\phi \theta(\phi) \right]_{\phi_{k-1}} \cdot \left[ \nabla_{\theta(\phi)} J(\pi_{\theta(\phi)}, \overline{R}) \right]_{\phi_{k-1}}$, where $\eta$ is the learning rate. We note that the LIRPG technique could fail in environments with extremely sparse rewards as the agent may not receive a non-zero extrinsic reward signal needed to update the parameter $\phi$. Moreover, the LIRPG technique is applicable only to policy-gradient based RL agents.

**Fully self-supervised reward shaping: SORS [26].** Self-supervised Online Reward Shaping (SORS) technique [26] considers a reward function of the form $\widehat{R}^{\text{SORS}}(s, a) = R_\phi(s, a)$, and infers the parameter $\phi$ using a classification-based reward inference algorithm, T-REX [27]. However, unlike T-REX that requires rankings over the trajectories as input, SORS uses the extrinsic reward $\overline{R}$ as a self-supervised learning signal to rank the trajectories generated by the agent during training. By de-

sign, SORS only enforces the relative pairwise ordering over the trajectories w.r.t. $\overline{R}$ when training $R_\phi$ and ignores the scale of the returns associated with trajectories w.r.t. $\overline{R}$. This makes training a policy challenging when the environment has noisy or distractive reward signals. Further, similar to LIRPG, the SORS technique could fail in environments with extremely sparse rewards as the agent may not obtain any trajectories with non-zero extrinsic reward signal needed to update the parameter $\phi$.

In this paper, we seek to develop an online reward shaping technique that can accelerate the agent's training process in environments with extremely sparse and distractive rewards, without any expert domain knowledge. As discussed above, techniques that rely only on intrinsic bonuses [32–34] could mislead the agent towards sub-optimal behaviors in noisy-distractive domains. Similarly, the fully self-supervised reward shaping techniques (LIRPG and SORS) might be ineffective in environments with extremely sparse rewards. We overcome these limitations by designing a novel reward shaping framework that appropriately balances exploration (via an intrinsic bonus component) and exploitation (via an intrinsic reward component) of extrinsic reward signals.

## 3 Exploration-Guided Reward Shaping

In Sections 3.1, 3.2, and 3.3, we propose an exploration-guided reward shaping framework, EXPLORS, to accelerate an RL agent's training process. In Section 3.4, we theoretically showcase the usefulness of our framework in a chain environment.

### 3.1 Our Reward Formulation

We consider the following parametric reward function for EXPLORS (see Algorithm 1):

$$\widehat{R}^{\text{EXPLORS}}(s,a) := \overline{R}(s,a) + R_\phi^{\text{SELFRS}}(s,a) + B_w^{\text{EXPLOB}}(s), \tag{1}$$

where $\phi \in \mathbb{R}^{d_\phi}$ and $w \in \mathbb{R}^{d_w}$. Here, $R_\phi^{\text{SELFRS}}$ corresponds to the intrinsic rewards in self-supervised reward shaping techniques, and $B_w^{\text{EXPLOB}}$ corresponds to the intrinsic bonuses in exploration-only reward shaping techniques. At round $k$ of Algorithm 1, $\widehat{R}_{k-1}^{\text{EXPLORS}}(s,a)$ is designed with parameters $(\phi_{k-1}, w_{k-1})$. Then, given updated policy $\pi_k$, we update the parameters $(\phi_{k-1}, w_{k-1})$ to $(\phi_k, w_k)$.

**Notation.** For the remainder of this section, we drop the superscripts (EXPLORS, SELFRS, and EXPLOB) when referring to the reward functions in Eq. (1). In the subscript of the expectations $\mathbb{E}$, let $\pi(a|s)$ mean $a \sim \pi(\cdot|s)$, $\mu^\pi(s,a)$ mean $s \sim d^\pi, a \sim \pi(\cdot|s)$, and $\mu^\pi(s)$ mean $s \sim d^\pi$. Further, we use shorthand notation $\mu_{s,a}^k$ and $\mu_s^k$ to refer $\mu^{\pi_{\theta_k}}(s,a)$ and $\mu^{\pi_{\theta_k}}(s)$, respectively.

**Intrinsic reward $R_\phi$.** We model the intrinsic reward $R_\phi$ using any parameterized function. At round $k$, for fixed $\pi_k$ and $w_{k-1}$, we update the parameter $\phi_{k-1}$ to $\phi_k$ by considering the effect such a change would have on the the expected cumulative return w.r.t. $\overline{R}$ through the change in the policy $\pi_k$ [24, 25]. In particular, we update $\phi$ using the gradient $\left[\nabla_\phi J(L(\pi_k, \widehat{R}), \overline{R})\right]_{\phi_{k-1}}$, where $\widehat{R}(s,a) = \overline{R}(s,a) + R_\phi(s,a) + B_{w_{k-1}}(s)$. However, when considering $L$ with neural policies, it is challenging to directly analyze the impact of $\phi$ in the policy $\pi_k$. Since our goal is to design a reward shaping technique that is applicable to any RL agent, we consider a simple surrogate learning algorithm $\widetilde{L}$ for our analysis. In particular, we consider $\widetilde{L}$ with parametric policies $\{\pi_\theta : \theta \in \mathbb{R}^{d_\theta}\}$ that does single-step vanilla policy gradient update with $Q$-values computed using $h$-depth planning. We map the policy $\pi_k$ to a parameter $\theta_k \in \mathbb{R}^{d_\theta}$ and define:

$$\widetilde{L}(\theta_k, \widehat{R}) := \theta_k + \alpha \cdot \left[\nabla_\theta J(\pi_\theta, \widehat{R})\right]_{\theta_k} = \theta_k + \alpha \cdot \mathbb{E}_{\mu_{s,a}^k}\left[\left[\nabla_\theta \log \pi_\theta(a|s)\right]_{\theta_k} Q_{\widehat{R},h}^{\pi_{\theta_k}}(s,a)\right],$$

where $\alpha$ is the learning rate and $Q_{\widehat{R},h}^{\pi_{\theta_k}}(s,a) = \mathbb{E}\left[\sum_{t=0}^h \gamma^t \widehat{R}(s_t, a_t)\big|s_0 = s, a_0 = a, T, \pi_{\theta_k}\right]$ is the $h$-depth $Q$-value w.r.t. $\widehat{R}$. Then, we update $\phi$ using the following bi-level optimization:

$$\arg\max_\phi \quad J(\pi_{\theta(\phi)}, \overline{R}) \tag{P1.U}$$

$$\text{subject to} \quad \theta(\phi) \leftarrow \widetilde{L}(\theta_k, \widehat{R}), \tag{P1.L}$$

where $\widehat{R}(s,a) := \overline{R}(s,a) + R_\phi(s,a) + B_{w_{k-1}}(s)$. In the above bi-level formulation, $\widetilde{L}$ with $h$-depth planning for small values of $h$ essentially requires designing more informative intrinsic rewards to benefit the agent's training process [24].

**Intrinsic bonus $B_w$.** Given a state abstraction $\psi : \mathcal{S} \to \mathcal{X}_\psi$ (with $|\mathcal{X}_\psi| = d_w$), we maintain the visitation count of the abstracted states in $w$, i.e., $w[x]$ corresponds to the visitation counts of the states $\{s \in \mathcal{S} : \psi(s) = x\}$. This allows us to implicitly maintain pseudo-counts $N_w(s)$ of visiting states $s \in \mathcal{S}$. In particular, we set $N_w(s) = \left(\frac{\lambda}{B_{\max}}\right)^2 + w[\psi(s)]$ for some $B_{\max}, \lambda > 0$. Then, we define the intrinsic bonus as follows: $B_w(s) = \frac{\lambda}{\sqrt{N_w(s)}}$. We update $w$ based on the rollouts in round $k$ [32–34].

## 3.2 Derivation of Gradient Updates for $R_\phi$

In this subsection, we first obtain high-level meta-gradient updates for $R_\phi$ similar to LIRPG [25]. Then, we derive intuitive meta-gradient updates that would allow EXPLORS to be compatible with any RL agent.

**High-level gradient updates for $R_\phi$.** We solve the bi-level optimization problem (P1.U)-(P1.L) of the intrinsic reward component in an iterative manner using the gradient updates that we derive below. At round $k$, for fixed $\pi_k$ and $w_{k-1}$, we update the parameter $\phi_{k-1}$ to $\phi_k$ as follows:

$$\phi_k = \phi_{k-1} + \eta \cdot \left[\nabla_\phi J(\pi_{\theta(\phi)}, \overline{R})\right]_{\phi_{k-1}} \overset{(a)}{=} \phi_{k-1} + \eta \cdot \left[\nabla_\phi \theta(\phi) \cdot \nabla_{\theta(\phi)} J(\pi_{\theta(\phi)}, \overline{R})\right]_{\phi_{k-1}}$$

$$\overset{(b)}{\approx} \phi_{k-1} + \eta \cdot \underbrace{\left[\nabla_\phi \theta(\phi)\right]_{\phi_{k-1}}}_{①} \cdot \underbrace{\left[\nabla_\theta J(\pi_\theta, \overline{R})\right]_{\theta_k}}_{②}, \quad (2)$$

where $\eta$ is the learning rate, the equality in $(a)$ is due to chain rule, and the approximation in $(b)$ is made by assuming a smoothness condition of $\left\| \left[\nabla_\theta J(\pi_\theta, \overline{R})\right]_{\theta(\phi_{k-1})} - \left[\nabla_\theta J(\pi_\theta, \overline{R})\right]_{\theta_k} \right\|_2 \leq c \cdot \|\theta(\phi_{k-1}) - \theta_k\|_2$ for some $c > 0$. By using the meta-gradient derivations in [47–49], we write the term ① as follows: $\left[\nabla_\phi \theta(\phi)\right]_{\phi_{k-1}} = \alpha \cdot \mathbb{E}_{\mu_{s,a}^k}\left[\left[\nabla_\phi Q_{\widehat{R},h}^{\pi_{\theta_k}}(s,a)\right]_{\phi_{k-1}} \cdot \left[\nabla_\theta \log \pi_\theta(a|s)\right]_{\theta_k}^\top\right]$, where $\widehat{R}(s,a) := \overline{R}(s,a) + R_\phi(s,a) + B_{w_{k-1}}(s)$. By using the policy gradient theorem [50], we write the term ② as follows: $\left[\nabla_\theta J(\pi_\theta, \overline{R})\right]_{\theta_k} = \mathbb{E}_{\mu_{s,a}^k}\left[\left[\nabla_\theta \log \pi_\theta(a|s)\right]_{\theta_k} Q_{\overline{R}}^{\pi_{\theta_k}}(s,a)\right]$. The above gradient update of $\phi_k$, involving the terms ① and ②, resembles the LIRPG [25] update. However, both the terms ① and ② require computing the gradient of the policy, i.e., $\nabla_\theta \log \pi_\theta(a|s)$. This requirement makes the above update applicable only for policy-gradient based agents. Below, we derive intuitive simplifications of the above two terms, ① and ②, that would enable our technique to be applicable to any RL agent, and not only policy-gradient based agents.

**Intuitive gradient updates for $R_\phi$.** In order to obtain intuitive forms of the terms ① and ②, we consider further simplifications to the surrogate learning algorithm $\widetilde{L}$ introduced in Section 3.1. In particular, for our analysis and derivation, we let $\widetilde{L}$ use tabular representation $\theta \in \mathbb{R}^{|\mathcal{S}| \cdot |\mathcal{A}|}$ and softmax policy given by $\pi_\theta(a|s) := \frac{\exp\left(\theta(s,a)\right)}{\sum_b \exp\left(\theta(s,b)\right)}, \forall s \in \mathcal{S}, a \in \mathcal{A}$. We define $A_{\widehat{R},h}^{\pi_{\theta_k}}(s,a) := Q_{\widehat{R},h}^{\pi_{\theta_k}}(s,a) - V_{\widehat{R},h}^{\pi_{\theta_k}}(s)$ and $A_{\overline{R}}^{\pi_{\theta_k}}(s,a) := Q_{\overline{R}}^{\pi_{\theta_k}}(s,a) - V_{\overline{R}}^{\pi_{\theta_k}}(s)$. Based on this, the following proposition provides intuitive gradient updates for $R_\phi$.

**Proposition 1.** *For the simplified surrogate learning algorithm $\widetilde{L}$ with $h$-depth planning, the gradient term $\left[\nabla_\phi \theta(\phi)\right]_{\phi_{k-1}} \cdot \left[\nabla_\theta J(\pi_\theta, \overline{R})\right]_{\theta_k}$ in Eq. (2) takes the following form:*

$$\alpha \cdot \mathbb{E}_{\mu_{s,a}^k}\left[\mu_{s,a}^k \cdot A_{\overline{R}}^{\pi_{\theta_k}}(s,a) \cdot \left[\nabla_\phi A_{\widehat{R},h}^{\pi_{\theta_k}}(s,a)\right]_{\phi_{k-1}}\right].$$

*For the special case of $h = 1$, the gradient term further simplifies to the following form:*

$$\alpha \cdot \mathbb{E}_{\mu_{s,a}^k}\left[\mu_{s,a}^k \cdot A_{\overline{R}}^{\pi_{\theta_k}}(s,a) \cdot \left[\nabla_\phi\big(R_\phi(s,a) - \mathbb{E}_{\pi_{\theta_k}(b|s)}[R_\phi(s,b)]\big)\right]_{\phi_{k-1}}\right].$$

Compared to Eq. (2), the intuitive gradient update term in the above proposition does not require computing the policy gradient $\nabla_\theta \log \pi_\theta(a|s)$. This allows us to develop an update rule for intrinsic reward parameter $\phi$ that is applicable to any RL agent. In particular, given the current policy $\pi_k$ (possibly without any differentiable parameterization), we simplify Eq. (2) and propose the following gradient update rule for parameter $\phi$:

$$\phi_k \approx \phi_{k-1} + \eta' \cdot \mathbb{E}_{\mu^k_{s,a}} \left[ \mu^k_{s,a} \cdot A^{\pi_k}_{\overline{R}}(s,a) \cdot \left[ \nabla_\phi \left( R_\phi(s,a) - \mathbb{E}_{\pi_k(b|s)}[R_\phi(s,b)] \right) \right]_{\phi_{k-1}} \right], \quad (3)$$

where $\eta' = \eta \cdot \alpha$. Note that the above gradient update only requires black-box access to the policy $\pi_k$ in the form of trajectory rollouts as in the SORS technique [26].

### 3.3 Empirical Updates and Practical Aspects

In this subsection, we present a concrete pseudocode for training an RL agent with EXPLORS reward shaping technique. Algorithm 2 provides a sketch of the overall training process, interleaving the agent's training with EXPLORS. The sketch presented in Algorithm 2 is adapted from the training process proposed for the SORS technique [26]. Further, we consider rollouts where each round corresponds to a single rollout, instead of environment steps, as in SORS. Below, we discuss the empirical updates for intrinsic reward and bonus components of Eq. (1).

**Empirical updates for intrinsic reward $R_\phi$.** We translate the final expectation-based update of $\phi_k$ in Eq. (3) to its empirical counterpart using the rollout data $\mathcal{D}$ collected by executing the current policy $\pi_k$ (or recent policies) in the MDP $M$. At any round $k$, let $\mathcal{D}$ contain a collection of trajectories $\{\xi^i\}_{i=1}^n$, where $\xi^i = (s_0^i, a_0^i, s_1^i, a_1^i, \dots, s_H^i)$. For a given trajectory $\xi^i$ and time index $t$, we denote a partial trajectory as $\xi_t^i = (s_t^i, a_t^i, \dots, s_H^i)$. Based on this notation, we empirically update the parameter $\phi$ as follows:

$$\phi_k \leftarrow \phi_{k-1} + \eta_k^\phi \cdot \sum_{\xi_t^i} \pi_k(a_t^i|s_t^i) \cdot \left( J(\xi_t^i, \overline{R}) - V^{\pi_k}_{\overline{R}}(s_t^i) \right) \cdot \left[ \nabla_\phi A^{\pi_k}_{\widehat{R},1}(s_t^i, a_t^i) \right]_{\phi_{k-1}}, \quad (4)$$

where we absorb the normalization factors into $\eta_k^\phi$, ignore the term $\mu^{\pi_k}(s_t^i)$, and set $\left[ \nabla_\phi A^{\pi_k}_{\widehat{R},1}(s_t^i, a_t^i) \right]_{\phi_{k-1}} = \left[ \nabla_\phi \left( R_\phi(s_t^i, a_t^i) - \mathbb{E}_{\pi_k(b|s_t^i)}[R_\phi(s_t^i, b)] \right) \right]_{\phi_{k-1}}$. Similar to LIRPG [25], we also maintain a critic $V_{\overline{R},\widetilde{\phi}_{k-1}}(\cdot)$ to approximate $V^{\pi_k}_{\overline{R}}(\cdot)$ in Eq. (4). We update the parameters of the critic, $\widetilde{\phi}_{k-1}$ to $\widetilde{\phi}_k$, using the same rollout data $\mathcal{D}$ and learning rate $\eta_k^{\widetilde{\phi}}$. In Algorithm 2, hyperparameters $N_r$ and $N_\pi$ control the frequency of updates for the intrinsic reward $R_\phi$ and policy $\pi$, respectively. For stability reasons, we update the policy more frequently compared to the intrinsic reward, i.e., $N_\pi < N_r$. We provide full implementation details in Section 4 and appendices.

**Empirical updates for intrinsic bonus $B_w$.** We update $B_w$ based on the history of all the states visited up to round $k$. Similar to #Exploration [34], we use the count-based intrinsic bonuses with a state abstraction $\psi : \mathcal{S} \to \mathcal{X}_\psi$. We maintain the visitation count of the abstracted states in $w$. For each rollout $\xi^k$, we update the parameter $w$ of the intrinsic bonus as follows:

$$w_k[x] = w_{k-1}[x] + \sum_{s_t^k \in \xi^k} \mathbf{1} \left\{ \psi(s_t^k) = x \right\}, \ \forall x \in \mathcal{X}_\psi. \quad (5)$$

Similar to the existing count-based exploration techniques [32–34], we use a lookahead step when incorporating the bonus term (see line 5 in Algorithm 2). In our implementation, we update the intrinsic bonus at a more fine-grained level, i.e., we update $B_w$ at each environment step $t$ within each round $k$ directly, instead of waiting for the rollout to finish. However, for clear presentation in Algorithm 2, we write the $B_w$ update at the level of round $k$, not at the level of environment step $t$. We provide full implementation details in Section 4 and appendices.

### 3.4 Theoretical Analysis

In this subsection, we theoretically showcase the usefulness of our exploration-guided reward shaping framework in accelerating an agent's learning in a chain environment with extremely-sparse rewards and distractive zones in the state space. Our analysis considers a stylized learning setting with simplified versions of different reward shaping techniques.

---

**Algorithm 2** RL Training with EXPLORS

1: **Inputs and hyperparameters:** RL algorithm $L$; first-in-first-out buffer $\mathcal{D}$ with size $D_{\max}$; abstraction $\psi$; learning rates $\{\eta_k^\phi\}$, $\{\eta_k^{\widetilde\phi}\}$; bonus parameters $B_{\max}$, $\lambda$; update rates $N_r$, $N_\pi$

2: **Initialization:** Initialize the parameters for intrinsic reward and its critic $(\phi_0, \widetilde\phi_0)$, parameters for intrinsic bonus $w_0$, and the policy $\pi_0$

3: **for** $k = 1, 2, \ldots, K$ **do**
    // policy update
4:    **if** $k \% N_\pi = 0$ **then**
5:        Define reward $\widehat{R}_{k-1}(s, a, s') := \overline{R}(s, a) + R_{\phi_{k-1}}(s, a) + B_{w_{k-1}}(s')$
6:        Obtain updated policy $\pi_k \leftarrow L(\pi_{k-1}, \widehat{R}_{k-1})$ using the latest rollouts in $\mathcal{D}$
7:    **else**
8:        Keep previous policy $\pi_k \leftarrow \pi_{k-1}$
    // data collection
9:    Rollout the policy $\pi_k$ in the MDP $M$ to obtain a trajectory $\xi^k = \left(s_0^k, a_0^k, s_1^k, a_1^k, \ldots, s_H^k\right)$
10:    Store $\xi^k$ in the buffer $\mathcal{D}.\mathrm{add}(\xi^k)$; if the buffer $\mathcal{D}$ is full, remove the oldest trajectory
    // intrinsic reward update
11:    **if** $k \% N_r = 0$ **then**
12:        Obtain updated reward parameter $\phi_k$ from $\phi_{k-1}$ as in Eq. (4) using $\mathcal{D}$ and learning rate $\eta_k^\phi$
13:        Obtain updated critic parameter $\widetilde\phi_k$ from $\widetilde\phi_{k-1}$ using $\mathcal{D}$ and learning rate $\eta_k^{\widetilde\phi}$
14:    **else**
15:        Keep previous parameters $\phi_k \leftarrow \phi_{k-1}$ and $\widetilde\phi_k \leftarrow \widetilde\phi_{k-1}$
    // intrinsic bonus update
16:    Update $w_k$ as in Eq. (5) using the states visited in the trajectory $\xi^k$
17:    Define bonus $B_{w_k}(s) = \frac{\lambda}{\sqrt{N_{w_k}(s)}}$, where $N_{w_k}(s) = \left(\frac{\lambda}{B_{\max}}\right)^2 + w_k[\psi(s)]$

18: **Output:** Policy $\pi_K$

---

**Chain environment.** We consider a chain environment $M = \left(\mathcal{S}, \mathcal{A}, T, P_0, \gamma, \overline{R}\right)$ of length $n_1 + n_2 + 1$. Let the state space be $\mathcal{S} = \{x_{-n_2}, \ldots, x_{-1}, x_0, x_1, \ldots, x_{n_1}\}$, and the action space be $\mathcal{A} = \{\leftarrow, \rightarrow\}$. We always start in the state $x_0$, i.e., the initial state distribution is $P_0(x_0) = 1$. The transition dynamics is deterministic and given as follows: $T(x_{i+1}|x_i, \rightarrow) = 1$ for $-n_2 \leq i \leq n_1 - 1$, $T(x_{i-1}|x_i, \leftarrow) = 1$ for $-(n_2 - 1) \leq i \leq n_1$, $T(\mathtt{terminal}|x_{n_1}, \rightarrow) = 1$, and $T(\mathtt{terminal}|x_{-n_2}, \leftarrow) = 1$. The reward function is defined as follows: $\overline{R}(x_i, \rightarrow) = 0$ for $-n_2 \leq i \leq n_1 - 1$, $\overline{R}(x_{n_1}, \rightarrow) = 1$, and $\overline{R}(x_i, \leftarrow) = 0$ for $-n_2 \leq i \leq n_1$. We consider an infinite horizon setting with discounted returns, i.e., $H \to \infty$ and $\gamma < 1$.

**Learning algorithm and reward shaping techniques.** For our theoretical analysis, we consider a stylized learning setting with a TD-style RL algorithm $L$ and simplified versions of different reward shaping techniques; details are provided in appendices. We analyze the total number time steps required for $L$ to learn an optimal policy in the chain environment under four different settings: (i) Case $L(\textsc{SelfRS} = 0, \textsc{ExploB} = 0)$ is a default setting without any shaping; (ii) Case $L(\textsc{SelfRS} = 0, \textsc{ExploB} = 1)$ uses only the intrinsic bonuses; (iii) Case $L(\textsc{SelfRS} = 1, \textsc{ExploB} = 0)$ uses only the intrinsic rewards; (iv) Case $L(\textsc{SelfRS} = 1, \textsc{ExploB} = 1)$ combines intrinsic bonuses with intrinsic rewards. The following theorem compares these four settings and showcases the usefulness of our framework, i.e., Case $L(\textsc{SelfRS} = 1, \textsc{ExploB} = 1)$ – proof is provided in appendices.

**Theorem 1.** *Consider the chain environment $M$ and the algorithm $L$ defined above. Let $cost(L(\textsc{SelfRS}, \textsc{ExploB}))$ denote the total number time steps required for $L(\textsc{SelfRS}, \textsc{ExploB})$ to learn an optimal policy in $M$. Then, we have the following (expected) costs for the four settings:*

*(i)* $\mathbb{E}\left[cost(L(\textsc{SelfRS} = 0, \textsc{ExploB} = 0))\right] \geq 2^{n_1 - 1}$;
*(ii)* $cost(L(\textsc{SelfRS} = 0, \textsc{ExploB} = 1)) = n_1 \cdot (n_1 + n_2 + 2)$;
*(iii)* $\mathbb{E}\left[cost(L(\textsc{SelfRS} = 1, \textsc{ExploB} = 0))\right] \geq 2^{n_1 - 1}$;
*(iv)* $cost(L(\textsc{SelfRS} = 1, \textsc{ExploB} = 1)) \leq n_1 + n_2 + 2$

The proof and additional details about the learning setting are provided in appendices.

# 4 Experimental Evaluation

In this section, we evaluate our reward shaping framework on three environments: CHAIN (Section 4.1), ROOM (Section 4.2), and LINEK (Section 4.3). CHAIN corresponds to a navigation task in a chain, adapted from the environment used for theoretical analysis in Section 3.4; this is a canonical environment used for studying extremely sparse-reward settings [7]. ROOM corresponds to a navigation task in a grid-world where the agent has to learn a policy to quickly reach the goal location in one of four rooms, starting from an initial location. Even though this environment has a small state/action space, it provides a very rich and intuitive problem setting to validate different reward shaping techniques. In fact, variants of ROOM have been used extensively in the literature [10, 11, 14, 17, 51–54]—the environment used in our experiments is adapted from [54]. LINEK corresponds to a navigation task in a one-dimensional space where the agent has to first pick the correct key and then reach the goal. The agent's location is represented as a point on a line segment. This environment is inspired by variants of navigation tasks in the literature where an agent needs to perform subtasks [3, 54, 55]—the environment used in our experiments is adapted from [54]. We give an overview of main results here, and provide a more detailed description of the setup and additional implementation details in appendices.

## 4.1 Evaluation on CHAIN

**CHAIN (Figure 1).** We represent the chain environment of length $n_1 + n_2 + 1$ as an MDP with state-space $\mathcal{S}$ consisting of an initial location $x_0$ (shown as "blue-circle"), $n_1$ nodes to the right of $x_0$, and $n_2$ nodes to the left of $x_0$. The rightmost

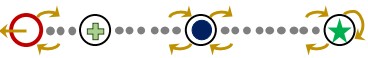

Figure 1: CHAIN$^0$ / CHAIN$^+$

node of the chain is the "goal" state (shown as "green-star"). In the left part of the chain, there can be a "distractor" state (shown as "green-plus"). The agent can take two actions given by $\mathcal{A} := \{\text{"left", "right"}\}$ – an action takes the agent to the intended neighboring node with probability of $(1 - p_{\text{rand}})$ for $p_{\text{rand}} = 0.05$. The agent receives a reward of $R_{\max} = 1$ for the "right" action at the goal state, $R_{\text{dis}}$ for the "left" action at the distractor state, and $0$ for all other state-action pairs. There is a discount factor $\gamma = 0.99$ and the environment resets after a horizon of $H = n_2$ steps. We consider two different variants of the chain environment: (i) CHAIN$^0$ with $(n_1 = 20, n_2 = 40, R_{\text{dis}} = 0)$; (ii) CHAIN$^+$ with $(n_1 = 20, n_2 = 40, R_{\text{dis}} = 0.01)$. We defer the full environment details to appendices.

**Evaluation setup.** We conduct our experiments with two different types of RL agents for CHAIN: tabular REINFORCE agent [7] and tabular Q-learning agent [7]. Algorithm 2 provides a sketch of the overall training process, and shows how agent's training interleaves with reward shaping techniques. We compare the performance of the following reward shaping techniques: (i) $\widehat{R}^{\text{ORIG}} := \overline{R}$ is a default baseline without any shaping; (ii) $\widehat{R}^{\text{SORS'}} := \overline{R} + R_\phi^{\text{SORS}}$ is based on the SORS technique [26] (see Section 2.2);[2] (iii) $\widehat{R}^{\text{LIRPG'}}$ is obtained via adapting the LIRPG technique [25] to our training pipeline (see Algorithm 2, Sections 2.2 and 3.2)—note that $\widehat{R}^{\text{LIRPG'}}$ is not applicable to Q-learning agent;[3] (iv) $\widehat{R}^{\text{EXPLOB}} := \overline{R} + B_w^{\text{EXPLOB}}$ uses only the intrinsic bonuses; (v) $\widehat{R}^{\text{SELFRS}} := \overline{R} + R_\phi^{\text{SELFRS}}$ uses only the intrinsic rewards; (vi) $\widehat{R}^{\text{EXPLORS}} := \overline{R} + R_\phi^{\text{SELFRS}} + B_w^{\text{EXPLOB}}$ combines intrinsic bonuses with intrinsic rewards. We provide full details about the implementation and hyperparameters in appendices.

**Results.** During training, the agent receives rewards based on $\widehat{R}$ and is evaluated based on $\overline{R}$. Figure 2 shows results for both the variants of CHAIN environment; the reported results are averaged over 20 runs and convergence plots show the mean performance with standard error bars. These results demonstrate the effectiveness of our exploration-guided reward shaping framework ($\widehat{R}^{\text{EXPLORS}}$), in comparison to baselines ($\widehat{R}^{\text{ORIG}}, \widehat{R}^{\text{SORS'}}, \widehat{R}^{\text{LIRPG'}}, \widehat{R}^{\text{EXPLOB}}, \widehat{R}^{\text{SELFRS}}$). Next, we summarize some of our key findings. First, our results show that $\widehat{R}^{\text{EXPLORS}}$ outperforms the baselines in both CHAIN$^0$ and CHAIN$^+$ environments, irrespective of the RL agent (REINFORCE and Q-learning). Second, the performance of $\widehat{R}^{\text{EXPLORS}}$ is better than variants which only use either intrinsic bonuses or intrinsic

---

[2]In our implementation, we use a variant of the SORS technique which also incorporates the extrinsic reward component $\overline{R}$ as done in all other techniques in our evaluation setup.

[3]Throughout the experimental evaluation, we refer to our implementation of the LIRPG technique as $\widehat{R}^{\text{LIRPG'}}$ instead of $\widehat{R}^{\text{LIRPG}}$ – our implementation of the LIRPG technique is not based on computing meta-gradients as in the original work [25]. Instead, we implemented $\widehat{R}^{\text{LIRPG'}}$ as a variant of $\widehat{R}^{\text{SELFRS}}$ where we set $h \to \infty$ instead of 1 in $A_{\widehat{R},h}^{\pi_{\theta_k}}(s, a)$ (see Section 3.2). We provide additional implementation details in appendices.

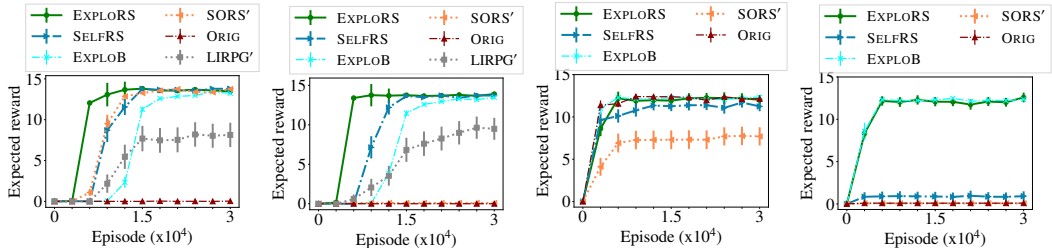

(a) CHAIN$^0$, REINFORCE    (b) CHAIN$^+$, REINFORCE    (c) CHAIN$^0$, Q-learning    (d) CHAIN$^+$, Q-learning

Figure 2: Results for CHAIN environment. These plots show convergence in performance of the agent w.r.t. training episodes. **(a, b)** show results for REINFORCE agent on CHAIN$^0$ (i.e., CHAIN variant without any distractor state) and CHAIN$^+$ (i.e., CHAIN variant with a distractor state). **(c, d)** show results for Q-learning agent on CHAIN$^0$ and CHAIN$^+$. See Section 4.1 for details.

rewards, i.e., $\widehat{R}^{\text{EXPLOB}}$ or $\widehat{R}^{\text{SELFRS}}$ – this demonstrates the utility of combining these two signals. Third, results in Figures 2b and 2d show that three reward shaping techniques ($\widehat{R}^{\text{SORS'}}$, $\widehat{R}^{\text{LIRPG'}}$, $\widehat{R}^{\text{SELFRS}}$) could fail or lead to sub-optimal policies because of the presence of distractor states.

## 4.2 Evaluation on ROOM

**ROOM (Figure 3).** This environment is based on the work of [54]; however, we adapted it to have a "distractor" state (shown as "green-plus") that provides a small reward. Similar to the two variants of CHAIN, we have two variants of this environment: (i) ROOM$^0$ has $R_{\text{dis}} = 0$ at the distractor state shown as "green-plus" (equivalently, there is no distractor state); (ii) ROOM$^+$ has $R_{\text{dis}} = 0.01$ at the distractor state. The environment-specific parameters (including $p_{\text{rand}}$, $R_{\text{max}}$, $\gamma$) are kept same as in Section 4.1. We defer full details to appendices.

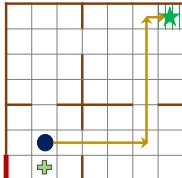

Figure 3: ROOM$^0$ / ROOM$^+$

**Evaluation setup and results.** Our evaluation setup for this environment is exactly same as that used for CHAIN environment (described in Section 4.1); here, we consider only the tabular REINFORCE agent. In particular, all the hyperparameters (related to the REINFORCE agent, reward shaping techniques, and training process) are the same as in Section 4.1. Figures 5a and 5b show the agent's performance for environments ROOM$^0$ and ROOM$^+$ (averaged over 20 runs). These results, along with results obtained in Figure 2, further demonstrate the effectiveness and robustness of $\widehat{R}^{\text{EXPLORS}}$ across different environments in comparison to baselines.

## 4.3 Evaluation on LINEK

**LINEK (Figure 4).** This environment corresponds to a navigation task in a one-dimensional space where the agent has to first pick the correct key and then reach the goal. The environment used in our experiments is based on the work of [54]; however, we adapted it to have multiple keys (only one being correct) and "distractor" states that provide a small reward at goal locations even without

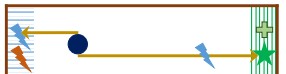

Figure 4: LINEK$^0$ / LINEK$^+$

the correct key. The environment comprises of the following main elements: (a) an agent whose current location (shown as "blue-circle") is a point x in $[0, 1]$; (b) goal (shown as "green-star") is available in locations on the segment $[0.9, 1]$; (c) a set of $k$ keys that are available in locations on the segment $[0.0, 0.1]$, (d) among $k$ keys, only 1 key is correct and the remaining $k - 1$ keys are wrong (i.e., irrelevant at the goal). Moreover, we consider the agent with two different actions related to picking a key: (a) "pickCorrect" makes the agent collect the correct key required at the goal; (b) "pickWrong" makes the agent collect one of the $k - 1$ wrong keys, chosen at random. Similar to Sections 4.1 and 4.2, we use two adaptations of the environment: (i) LINEK$^0$ with $(k = 10, R_{\text{dis}} = 0)$; (ii) LINEK$^+$ with $(k = 10, R_{\text{dis}} = 0.01)$. We defer full details to appendices.

**Experimental setup.** We conduct our experiments with a neural REINFORCE agent using a two-layered neural network architecture (i.e., one fully connected hidden layer with 256 nodes and RELU activation) [7]. Similar to Section 4.1, we compare the performance of six techniques. As a crucial difference, here we use neural-network based reward functions for $\widehat{R}^{\text{SORS'}}$, $\widehat{R}^{\text{LIRPG'}}$, $\widehat{R}^{\text{SELFRS}}$, and

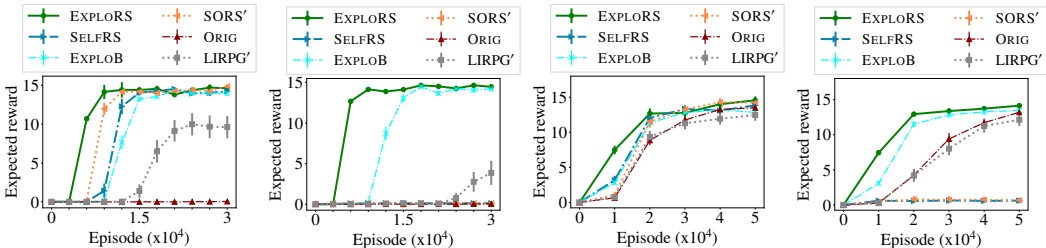

(a) ROOM$^0$, REINFORCE (b) ROOM$^+$, REINFORCE (c) LINEK$^0$, REINFORCE (d) LINEK$^+$, REINFORCE

Figure 5: Results for ROOM and LINEK environments. These plots show convergence in performance of the agent w.r.t. training episodes. **(a, b)** show results for REINFORCE agent on ROOM$^0$ (i.e., ROOM variant without any distractor state) and ROOM$^+$ (i.e., ROOM variant with a distractor state). **(c, d)** show results for REINFORCE agent on LINEK$^0$ (i.e., LINEK variant without any distractor state) and LINEK$^+$ (i.e., LINEK variant with distractor states). See Sections 4.2 and 4.3 for details.

$\widehat{R}^{\text{EXPLORS}}$ (see Footnotes 2 and 3). Based on [25, 26], we use the same neural-network architecture for intrinsic reward functions as used for the agent's policy by applying appropriate transformations at the output layer (e.g., instead of using soft-max, use $tanh$-clipping to get output reward values for actions). We provide full details about the implementation and hyperparameters in appendices.

**Results.** During training, the agent receives rewards based on $\widehat{R}$ and is evaluated based on $\overline{R}$. Figures 5c and 5d show results for both the variants of LINEK environment; the reported results are averaged over 30 runs and convergence plots show the mean performance with standard error bars. These plots showcase the performance of different techniques as we vary $R_{\text{dis}} \in \{0.00, 0.01\}$ – this in turn decides whether there are any distractor states that can serve as local minima for the agent. The convergence behavior in Figures 5c and 5d demonstrates the effectiveness of our exploration-guided reward shaping framework ($\widehat{R}^{\text{EXPLORS}}$), in comparison to baselines ($\widehat{R}^{\text{ORIG}}$, $\widehat{R}^{\text{SORS'}}$, $\widehat{R}^{\text{LIRPG'}}$, $\widehat{R}^{\text{EXPLOB}}$, $\widehat{R}^{\text{SELFRS}}$). Next, we summarize some of our key findings. First, our results show that $\widehat{R}^{\text{EXPLORS}}$ outperforms all the baselines in both LINEK$^0$ and LINEK$^+$ environments. Second, results in Figure 5d show that three reward shaping techniques ($\widehat{R}^{\text{SORS'}}$, $\widehat{R}^{\text{LIRPG'}}$, $\widehat{R}^{\text{SELFRS}}$) performed worse than $\widehat{R}^{\text{ORIG}}$ – this is because of the presence of distractor states which create local minima for the agent and these shaped functions could further encourage learning a sub-optimal policy. In contrast, $\widehat{R}^{\text{EXPLORS}}$ combines the benefits of intrinsic rewards ($\widehat{R}^{\text{SELFRS}}$) and intrinsic bonuses ($\widehat{R}^{\text{EXPLOB}}$) to speed up agent's learning in a robust and efficient manner. Overall, these results demonstrate that our shaping technique $\widehat{R}^{\text{EXPLORS}}$ results in efficient learning even when dealing with complex state representations and when learning neural-network based intrinsic reward functions.

## 5 Concluding Discussions

We proposed a novel reward shaping framework, EXPLORS, that operates in a fully self-supervised manner and could accelerate an agent's learning even in sparse-reward environments. Next, we discuss a few limitations of our work and outline a future plan to address them. First, the experimental evaluation is conducted on simpler environments to study the performance of techniques w.r.t. the three characteristics of (a) hard exploration, (b) local minima, and (c) "noisy TV" problem. It would be interesting to evaluate different reward design techniques in more complex environments (e.g., with continuous state/action spaces); this would also require designing benchmark environments that systematically capture the above three characteristics. Second, EXPLORS combines the intrinsic rewards and intrinsic bonuses that allows it to overcome the limitations of state-of-the-art techniques. It would be interesting to develop more principled ways to combine these two signals. Third, it would be useful to provide rigorous analysis of EXPLORS in terms of convergence speed and stability of an agent.

## 6 Acknowledgments

Parameswaran Kamalaruban acknowledges support from The Alan Turing Institute. Funded/Co-funded by the European Union (ERC, TOPS, 101039090). Views and opinions expressed are however those of the author(s) only and do not necessarily reflect those of the European Union or the European Research Council. Neither the European Union nor the granting authority can be held responsible for them.

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
