## A   Table of Contents

In this section, we give a brief description of the content provided in the appendices of the paper.

## B   Derivation of Gradient Updates for $R_\phi$: Proof (Section 3.2)

*Proof of Proposition 1.* For any $s \in \mathcal{S}, a \in \mathcal{A}$, let $\mathbf{1}_{s,a} \in \mathbb{R}^{|\mathcal{S}| \cdot |\mathcal{A}|}$ denote a vector with 1 in the $(s,a)$-th entry and 0 elsewhere. First, we simplify the term ① as follows:

$$
\frac{1}{\alpha} \cdot [\nabla_\phi \theta(\phi)]_{\phi_{k-1}}
$$

$$
= \mathbb{E}_{\mu_{s,a}^k} \left[ \left[ \nabla_\phi Q_{\widehat{R},h}^{\pi_{\theta_k}}(s,a) \right]_{\phi_{k-1}} \cdot \left[ \nabla_\theta \log \pi_\theta(a|s) \right]_{\theta_k}^\top \right]
$$

$$
= \mathbb{E}_{\mu_{s,a}^k} \left[ \left[ \nabla_\phi Q_{\widehat{R},h}^{\pi_{\theta_k}}(s,a) \right]_{\phi_{k-1}} \cdot \left( \mathbf{1}_{s,a} - \sum_{a'} \pi_{\theta_k}(a'|s) \cdot \mathbf{1}_{s,a'} \right)^\top \right]
$$

$$
= \mathbb{E}_{\mu_s^k} \left[ \sum_a \pi_{\theta_k}(a|s) \cdot \left[ \nabla_\phi Q_{\widehat{R},h}^{\pi_{\theta_k}}(s,a) \right]_{\phi_{k-1}} \cdot \left( \mathbf{1}_{s,a} - \sum_{a'} \pi_{\theta_k}(a'|s) \cdot \mathbf{1}_{s,a'} \right)^\top \right]
$$

$$
= \mathbb{E}_{\mu_s^k} \left[ \sum_a \pi_{\theta_k}(a|s) \left[ \nabla_\phi Q_{\widehat{R},h}^{\pi_{\theta_k}}(s,a) \right]_{\phi_{k-1}} \mathbf{1}_{s,a}^\top - \sum_a \pi_{\theta_k}(a|s) \left[ \nabla_\phi Q_{\widehat{R},h}^{\pi_{\theta_k}}(s,a) \right]_{\phi_{k-1}} \left( \sum_{a'} \pi_{\theta_k}(a'|s) \mathbf{1}_{s,a'}^\top \right) \right]
$$

$$
= \mathbb{E}_{\mu_s^k} \left[ \sum_a \pi_{\theta_k}(a|s) \left[ \nabla_\phi Q_{\widehat{R},h}^{\pi_{\theta_k}}(s,a) \right]_{\phi_{k-1}} \mathbf{1}_{s,a}^\top - \left[ \nabla_\phi \sum_a \pi_{\theta_k}(a|s) Q_{\widehat{R},h}^{\pi_{\theta_k}}(s,a) \right]_{\phi_{k-1}} \left( \sum_{a'} \pi_{\theta_k}(a'|s) \mathbf{1}_{s,a'}^\top \right) \right]
$$

$$
= \mathbb{E}_{\mu_s^k} \left[ \sum_a \pi_{\theta_k}(a|s) \cdot \left[ \nabla_\phi Q_{\widehat{R},h}^{\pi_{\theta_k}}(s,a) \right]_{\phi_{k-1}} \cdot \mathbf{1}_{s,a}^\top - \left[ \nabla_\phi V_{\widehat{R},h}^{\pi_{\theta_k}}(s) \right]_{\phi_{k-1}} \cdot \left( \sum_{a'} \pi_{\theta_k}(a'|s) \cdot \mathbf{1}_{s,a'} \right)^\top \right]
$$

$$
= \mathbb{E}_{\mu_s^k} \left[ \sum_a \pi_{\theta_k}(a|s) \cdot \left[ \nabla_\phi Q_{\widehat{R},h}^{\pi_{\theta_k}}(s,a) \right]_{\phi_{k-1}} \cdot \mathbf{1}_{s,a}^\top - \left[ \nabla_\phi V_{\widehat{R},h}^{\pi_{\theta_k}}(s) \right]_{\phi_{k-1}} \cdot \left( \sum_a \pi_{\theta_k}(a|s) \cdot \mathbf{1}_{s,a}^\top \right) \right]
$$

$$
= \mathbb{E}_{\mu_s^k} \left[ \sum_a \pi_{\theta_k}(a|s) \cdot \left[ \nabla_\phi Q_{\widehat{R},h}^{\pi_{\theta_k}}(s,a) \right]_{\phi_{k-1}} \cdot \mathbf{1}_{s,a}^\top - \sum_a \pi_{\theta_k}(a|s) \cdot \left[ \nabla_\phi V_{\widehat{R},h}^{\pi_{\theta_k}}(s) \right]_{\phi_{k-1}} \cdot \mathbf{1}_{s,a}^\top \right]
$$

$$
= \mathbb{E}_{\mu_{s,a}^k} \left[ \left[ \nabla_\phi Q_{\widehat{R},h}^{\pi_{\theta_k}}(s,a) \right]_{\phi_{k-1}} \cdot \mathbf{1}_{s,a}^\top - \left[ \nabla_\phi V_{\widehat{R},h}^{\pi_{\theta_k}}(s) \right]_{\phi_{k-1}} \cdot \mathbf{1}_{s,a}^\top \right]
$$

$$
= \mathbb{E}_{\mu_{s,a}^k} \left[ \left[ \nabla_\phi \left( Q_{\widehat{R},h}^{\pi_{\theta_k}}(s,a) - V_{\widehat{R},h}^{\pi_{\theta_k}}(s) \right) \right]_{\phi_{k-1}} \cdot \mathbf{1}_{s,a}^\top \right].
$$

Then, we simplify the term ② as follows:

$$
\left[ \nabla_\theta J(\pi_\theta, \overline{R}) \right]_{\theta_k}
$$

$$
= \mathbb{E}_{\mu_{s,a}^k} \left[ \left[ \nabla_\theta \log \pi_\theta(a|s) \right]_{\theta_k} \cdot Q_{\overline{R}}^{\pi_{\theta_k}}(s,a) \right]
$$

$$
= \mathbb{E}_{\mu_{s,a}^k} \left[ \left( \mathbf{1}_{s,a} - \sum_{a'} \pi_{\theta_k}(a'|s) \cdot \mathbf{1}_{s,a'} \right) \cdot Q_{\overline{R}}^{\pi_{\theta_k}}(s,a) \right]
$$

$$
= \mathbb{E}_{\mu_s^k} \left[ \sum_a \pi_{\theta_k}(a|s) \cdot \left( \mathbf{1}_{s,a} - \sum_{a'} \pi_{\theta_k}(a'|s) \cdot \mathbf{1}_{s,a'} \right) \cdot Q_{\overline{R}}^{\pi_{\theta_k}}(s,a) \right]
$$

$$= \mathbb{E}_{\mu_s^k}\left[\sum_a \pi_{\theta_k}(a|s) \cdot Q_{\overline{R}}^{\pi_{\theta_k}}(s,a) \cdot \mathbf{1}_{s,a} - \sum_a \pi_{\theta_k}(a|s) \cdot Q_{\overline{R}}^{\pi_{\theta_k}}(s,a) \cdot \left(\sum_{a'} \pi_{\theta_k}(a'|s) \cdot \mathbf{1}_{s,a'}\right)\right]$$

$$= \mathbb{E}_{\mu_s^k}\left[\sum_a \pi_{\theta_k}(a|s) \cdot Q_{\overline{R}}^{\pi_{\theta_k}}(s,a) \cdot \mathbf{1}_{s,a} - V_{\overline{R}}^{\pi_{\theta_k}}(s) \cdot \left(\sum_{a'} \pi_{\theta_k}(a'|s) \cdot \mathbf{1}_{s,a'}\right)\right]$$

$$= \mathbb{E}_{\mu_s^k}\left[\sum_a \pi_{\theta_k}(a|s) \cdot Q_{\overline{R}}^{\pi_{\theta_k}}(s,a) \cdot \mathbf{1}_{s,a} - \sum_{a'} \pi_{\theta_k}(a'|s) \cdot V_{\overline{R}}^{\pi_{\theta_k}}(s) \cdot \mathbf{1}_{s,a'}\right]$$

$$= \mathbb{E}_{\mu_s^k}\left[\sum_a \pi_{\theta_k}(a|s) \cdot Q_{\overline{R}}^{\pi_{\theta_k}}(s,a) \cdot \mathbf{1}_{s,a} - \sum_a \pi_{\theta_k}(a|s) \cdot V_{\overline{R}}^{\pi_{\theta_k}}(s) \cdot \mathbf{1}_{s,a}\right]$$

$$= \mathbb{E}_{\mu_s^k}\left[\sum_a \pi_{\theta_k}(a|s) \cdot \left(Q_{\overline{R}}^{\pi_{\theta_k}}(s,a) - V_{\overline{R}}^{\pi_{\theta_k}}(s)\right) \cdot \mathbf{1}_{s,a}\right]$$

$$= \mathbb{E}_{\mu_{s,a}^k}\left[\left(Q_{\overline{R}}^{\pi_{\theta_k}}(s,a) - V_{\overline{R}}^{\pi_{\theta_k}}(s)\right) \cdot \mathbf{1}_{s,a}\right].$$

Finally, we consider the following:

$$\left[\nabla_\phi \theta(\phi)\right]_{\phi_{k-1}} \cdot \left[\nabla_\theta J(\pi_\theta, \overline{R})\right]_{\theta_k}$$

$$= \alpha \cdot \mathbb{E}_{\mu_{s,a}^k}\left[\left[\nabla_\phi A_{\widehat{R},h}^{\pi_{\theta_k}}(s,a)\right]_{\phi_{k-1}} \cdot \mathbf{1}_{s,a}^\top\right] \cdot \mathbb{E}_{\mu_{s',a'}^k}\left[A_{\overline{R}}^{\pi_{\theta_k}}(s',a') \cdot \mathbf{1}_{s',a'}\right]$$

$$= \alpha \cdot \mathbb{E}_{\mu_{s,a}^k}\left[\left[\nabla_\phi A_{\widehat{R},h}^{\pi_{\theta_k}}(s,a)\right]_{\phi_{k-1}} \cdot \mathbf{1}_{s,a}^\top \cdot \mathbb{E}_{\mu_{s',a'}^k}\left[A_{\overline{R}}^{\pi_{\theta_k}}(s',a') \cdot \mathbf{1}_{s',a'}\right]\right]$$

$$= \alpha \cdot \mathbb{E}_{\mu_{s,a}^k}\left[\left[\nabla_\phi A_{\widehat{R},h}^{\pi_{\theta_k}}(s,a)\right]_{\phi_{k-1}} \cdot \mathbf{1}_{s,a}^\top \cdot \mu_{s,a}^k \cdot A_{\overline{R}}^{\pi_{\theta_k}}(s,a) \cdot \mathbf{1}_{s,a}\right]$$

$$= \alpha \cdot \mathbb{E}_{\mu_{s,a}^k}\left[\mu_{s,a}^k \cdot A_{\overline{R}}^{\pi_{\theta_k}}(s,a) \cdot \left[\nabla_\phi A_{\widehat{R},h}^{\pi_{\theta_k}}(s,a)\right]_{\phi_{k-1}}\right]$$

$$= \alpha \cdot \mathbb{E}_{\mu^{\pi_{\theta_k}}(s,a)}\left[\mu^{\pi_{\theta_k}}(s) \cdot \pi_{\theta_k}(a|s) \cdot A_{\overline{R}}^{\pi_{\theta_k}}(s,a) \cdot \left[\nabla_\phi A_{\widehat{R},h}^{\pi_{\theta_k}}(s,a)\right]_{\phi_{k-1}}\right].$$

$\square$

## C   Theoretical Analysis: Proof (Section 3.4)

*Proof of Theorem 1.* We prove Theorem 1 via case-by-case analysis.

**Case** $L(\text{SELFRS} = 0, \text{EXPLOB} = 0)$. This case corresponds to learning without any reward shaping, i.e., learning with the extrinsic reward only: $\overline{R}(s,a)$. Then, we note the following:

I.  Initially, we have a random policy except at state $x_{n_1}$, where we take the optimal action $\rightarrow$ (line 7). We maintain zero value function $V_t$ for all the states (line 9) until we obtain the first success complete rollout, i.e., $s_{t+1}$ is terminal and $\overline{R}(s_t, a_t) = 1$.

II. With an initial random policy and starting from $x_0$, probability of obtaining a success complete rollout is $\left(\frac{1}{2}\right)^{n_1} + \left(\frac{1}{2}\right)^{n_1+2} + \left(\frac{1}{2}\right)^{n_1+4} + \dots$, which is upper bounded by $p_{\max} = \sum_{i=0}^\infty \left(\frac{1}{2}\right)^{n_1+i} = \left(\frac{1}{2}\right)^{n_1-1}$.

III. Let $\mathbb{E}[T_1]$ be the expected number of steps required for the first occurrence of the above successful rollout. Then, we have: $\mathbb{E}[T_1] \geq \frac{1}{p_{\max}} = 2^{n_1-1}$.

IV. After the first successful rollout, we will have $V_t(x_{n_1}) = 1$ and zero elsewhere (line 9). Then, we will have a random policy except at $x_{n_1}$ and $x_{(n_1-1)}$, where we take the optimal action (line 7). This effectively repeats the same steps above for the chain without $x_{n_1}$.

V.  Let $\mathbb{E}[T_2]$ be the expected number of steps required for the second occurrence of the above successful rollout. Then, we have: $\mathbb{E}[T_2] \geq 2^{n_1-2}$.

---

**Algorithm 3** Simplified RL Algorithm $L$ with Reward Shaping

---

1: **Input:** Binary flags SELFRS and EXPLOB
2: **Initialize:** $V_0(s) = 0$; $R(s,a) = 0$, $B(s) = 1$, $\forall s \in \mathcal{S}, a \in \mathcal{A}$; $\lambda \in (0,1)$
3: $s_1 = x_0$; $B(s_1) = \lambda$
4: **for** each $t = 1, 2, \ldots$ **do**
5:     **if** EXPLOB $= 0$ **then**
6:         $B(s) = 0, \forall s \in \mathcal{S}$
        // bonus component used for action selection
7:     $a_t = \arg\max_{a'} \overline{R}(s_t, a') + R(s_t, a') + B(T(s_t, a')) + \gamma \cdot V_{t-1}(T(s_t, a'))$
8:     $s_{t+1} = T(s_t, a_t)$
    // we do not consider the bonus component when updating the value function
9:     $V_t(s_t) = \overline{R}(s_t, a_t) + R(s_t, a_t) + \gamma \cdot V_{t-1}(s_{t+1})$
10:     **if** $s_{t+1} = $ terminal **then**
11:         **if** $\overline{R}(s_t, a_t) = 1$ and SELFRS $= 1$ **then**
            // update the intrinsic reward component
12:             $\phi(s) = 0, \forall s \in \mathcal{S}$
13:             Update $\phi(s)$ for all the states in the current rollout as the discounted return
14:             $R(s,a) = \gamma \cdot \phi(T(s,a)) - \phi(s), \forall s \in \mathcal{S}, a \in \mathcal{A}$
            // reset the value function to account for change in $R$
15:             $V_t(s) = 0, \forall s \in \mathcal{S}$
16:         reset $s_{t+1} = x_0$
    // update the bonus component
17:     $B(s_{t+1}) = \lambda \cdot B(s_{t+1})$
18: **Output:** policy $\pi_t$

---

VI. After the second successful rollout, we will have $V_t(x_{n_1}) = 1$, $V_t(x_{(n_1-1)}) = \gamma$, and zero elsewhere (line 9). Then, we will have a random policy except at $x_{n_1}$, $x_{(n_1-1)}$, and $x_{(n_1-2)}$, where we take the optimal action (line 7). This effectively repeats the same steps above for the chain without $x_{n_1}$ and $x_{(n_1-1)}$.

VII. After following the above procedure for $n_1$ success rollouts, we will have the optimal value/policy learnt for the chain (solving the MDP). Thus, the expected sample complexity is lower bounded by $\mathbb{E}\left[\text{cost}(L(\text{SELFRS} = 0, \text{EXPLOB} = 0))\right] = \sum_{i=1}^{n_1} \mathbb{E}[T_i] \geq \sum_{i=1}^{n_1} 2^{n_1 - i}$.

**Case** $L(\text{SELFRS} = 0, \text{EXPLOB} = 1)$ This case corresponds to learning with the extrinsic reward and intrinsic bonus: $\overline{R}(s,a) + B(T(s,a))$. Then, we note the following (here, we need $\lambda \leq \gamma$):

I. We have zero value function (line 9) until we get the first success complete rollout, i.e., $s_{t+1}$ is terminal and $\overline{R}(s_t, a_t) = 1$.

II. W.l.o.g. we take $\rightarrow$ action at time $t = 1$ at $x_0$. Then, we continue to take $\rightarrow$ action (for $n_1 + 1$ steps) until we reach rightmost terminal state, since $\lambda < 1$ (lines 7 and 17).

III. After the first successful rollout, we will have $V_t(x_{n_1}) = 1$ and zero elsewhere (line 9). Note that $V_t(\text{terminal}) = 0, \forall t$.

IV. Once we reset to $x_0$, we take $\leftarrow$ since $\lambda < 1$ (line 7). Then, we continue to take $\leftarrow$ action (for $n_2 + 1$ steps) until we reach leftmost terminal state, since $\lambda < 1$ (lines 7 and 17).

V. This alternating one-sided navigation process will continue until $V_t$ values are updated for all the nodes right to $x_0$ (one node at a time per one full cycle). The condition $\lambda \leq \gamma$ ensures that after all the nodes right to $x_0$ get updated with right $V_t$ values, there will be no further exploration on the left-side of $x_0$. Thus, the sample complexity is given by $\text{cost}(L(\text{SELFRS} = 0, \text{EXPLOB} = 1)) = n_1 \cdot (n_1 + n_2 + 2)$.

**Case** $L(\text{SELFRS} = 1, \text{EXPLOB} = 0)$ This case corresponds to learning with the extrinsic reward and intrinsic reward : $\overline{R}(s,a) + R(s,a)$. Then, we note the following:

I. From the analysis for the case $L(1, 1)$, we have: $\mathbb{E}[T_1] \geq \frac{1}{p_{\max}} = 2^{n_1 - 1}$.

II. However, after the first successful rollout, we obtain the optimal policy (line 7) immediately since the shaping reward (line 14) contains myopic-optimality information. Thus, the expected sample complexity is lower bounded by $\mathbb{E}[\text{cost}(L(\text{SELFRS} = 1, \text{EXPLOB} = 0))] = \mathbb{E}[T_1] \geq 2^{n_1 - 1}$.

**Case** $L(\text{SELFRS} = 1, \text{EXPLOB} = 1)$ This case corresponds to learning with the extrinsic reward and intrinsic reward and bonus: $\overline{R}(s, a) + R(s, a) + B(T(s, a))$. Then, we note the following (here, we need $\lambda^2 \leq \gamma^{n_1}$):

I. From the analysis for the case $L(1, 0)$, we obtain first successful trajectory after $n_1 + n_2 + 2$ steps (utmost). Then, as in the case of $L(0, 1)$, shaping reward (line 14) will propagate myopic-optimality information immediately. The condition $\lambda^2 \leq \gamma^{n_1}$ ensures that after all the nodes right to $x_0$ get updated with right $V_t$ values, there will be no further exploration on the left-side of $x_0$. Thus, the sample complexity is upper bounded by $\text{cost}(L(\text{SELFRS} = 1, \text{EXPLOB} = 1)) \leq n_1 + n_2 + 2$.

$\square$

# D    Evaluation on CHAIN: Additional Details (Section 4.1)

**CHAIN (Figure 1).** We expand on the details of the CHAIN environment, introduced in Section 4.1. We represent the chain environment of length $n_1 + n_2 + 1$ as an MDP with state-space $\mathcal{S}$ consisting of an initial location $x_0$ (shown as "blue-circle"), $n_1$ nodes to the right of $x_0$, and $n_2$ nodes to the left of $x_0$. The rightmost node of the chain is the "goal" state (shown as "green-star"). In the left part of the chain, there can be a "distractor" state (shown as "green-plus"). The agent can take two actions given by $\mathcal{A} := \{\text{"left"}, \text{"right"}\}$. An action takes the agent to the neighboring node represented by the direction of the action. However, taking "left" action at the leftmost node (shown as "thick-red-circle") leads to termination, and "right" action at the rightmost node (goal) keeps the agent at the current location. Furthermore, when an agent takes an action $a \in \mathcal{A}$, there is $p_{\text{rand}}$ probability that an action $a' \in \mathcal{A} \setminus \{a\}$ will be executed instead of $a$. The agent receives rewards as follows: $R_{\max}$ for the "right" action at the goal state, $R_{\text{dis}}$ for the "left" action at the distractor state, and 0 for all other state-action pairs. There is a discount factor $\gamma$ and the environment resets after a horizon of $H = n_2$ steps. In our evaluation, we set $p_{\text{rand}} = 0.05$, $R_{\max} = 1$, $R_{\text{dis}} = 0$ or 0.01, and $\gamma = 0.99$. We obtain different variants of the chain environment by changing the values of $(n_1, n_2, R_{\text{dis}})$. We consider two different variants of the chain environment: (i) CHAIN$^0$ with $(n_1 = 20, n_2 = 40, R_{\text{dis}} = 0)$; (ii) CHAIN$^+$ with $(n_1 = 20, n_2 = 40, R_{\text{dis}} = 0.01)$. The "distractor" state (shown as "green-plus") with $R_{\text{dis}}$ reward is located 15 nodes to the left of $x_0$ in both the environments.

**Evaluation setup: agents.** As mentioned in Section 4.1, we conduct our experiments with two different types of RL agents for CHAIN: tabular REINFORCE agent [7] and tabular Q-learning agent [7]. First, we consider tabular REINFORCE agent that maintain scores $\theta[s, a]$ for each state-action pair and applies soft-max operation over the scores to obtain the policy $\pi$. When computing the agent's performance during evaluation, we also use the agent's soft-max policy (instead of choosing actions greedily). Second, we consider tabular Q-learning agent with exploration factor $\epsilon = 0.05$. When computing the agent's performance during evaluation, we also use the agent's $\epsilon$-greedy policy (instead of choosing actions greedily). Algorithm 2 provides a sketch of the overall training process, and shows how agent's training interleaves with reward shaping techniques – the agent's policy is updated in lines 4–8 of the algorithm. For the agent's training process, we use a fixed set of hyperparameters irrespective of the type of agent or the reward shaping technique. More concretely, we have the following: (a) the agent's learning rate is set to 0.1; (b) frequency of updates $N_\pi$ is set to be 2, i.e., update after every 2 rollouts in the environment; (c) a rollout buffer (first-in-first-out) $\mathcal{D}$ of size 10 is maintained and we update the agent's policy using the last 5 rollouts in $\mathcal{D}$. In the tabular setting with CHAIN, we find that the overall quantitative results are robust to these hyperparameters – we use the exact same set of hyperparameters for evaluation on ROOM, described in Section 4.2.

**Evaluation setup: shaping techniques.** Next, we describe different reward shaping techniques used during the agent's training phase. Specifically, during training, the agent receives rewards based on

the shaped reward function $\widehat{R}$; the performance (as reported in the plots) is always evaluated w.r.t. the extrinsic reward function $\overline{R}$. More concretely, we have the following shaping techniques:

- $\widehat{R}^{\text{ORIG}} := \overline{R}$. This serves as a default baseline where extrinsic reward function is used during training without any shaping.

- $\widehat{R}^{\text{SORS'}} := \overline{R} + R_\phi^{\text{SORS}}$. This is based on the SORS technique[26]; see additional details in Section 2.2 (also see Footnote 2 about $\widehat{R}^{\text{SORS'}}$). For CHAIN environment, we use tabular representation for $R_\phi^{\text{SORS}}$ and perform gradient updates as described in the work of [26]. Algorithm 2 provides a sketch of the overall training process – the $R_\phi^{\text{SORS}}$ updates would be applied in lines 11–15 in the algorithm. In fact, the training process presented in Algorithm 2 is adapted from the training process proposed for the SORS technique [26]. We update the intrinsic reward function using the following hyperparameters: (a) the learning rate is set to $0.01$; (b) frequency of updates $N_r$ is set to be 5, i.e., update after every 5 rollouts in the environment; (c) we have a rollout buffer $\mathcal{D}$ of size 10 and sample a set of 10 *pairs* of rollouts for the gradient updates (in our implementation, we prioritized sampling of pairs that have non-zero gap between returns).

- $\widehat{R}^{\text{LIRPG'}} := \overline{R} + R_\phi^{\text{LIRPG'}}$. This is obtained via adapting the LIRPG technique of [25] to our training pipeline; see Algorithm 2, Sections 2.2 and 3.2 (also see Footnote 3 about $\widehat{R}^{\text{LIRPG'}}$). More specifically, when considering tabular REINFORCE agent, we implemented $\widehat{R}^{\text{LIRPG'}}$ as an adaptation of $\widehat{R}^{\text{SELFRS}}$ where we set $h \to \infty$ instead of 1 (see Section 3.2) – the rest of the implementation is same as described below for $\widehat{R}^{\text{SELFRS}}$. Note that the LIRPG technique is not applicable to Q-learning agent.

- $\widehat{R}^{\text{EXPLOB}} := \overline{R} + B_w^{\text{EXPLOB}}$. This corresponds to a part of our reward shaping technique which uses only the intrinsic bonuses $B_w^{\text{EXPLOB}}$. As discussed in Sections 3.1 and 3.3, we use a count-based bonus $B_w^{\text{EXPLOB}}$. For CHAIN environment, we use a tabular representation for $B_w^{\text{EXPLOB}}$ where $w[s]$ captures the state-visitation counts for a state $s$. Algorithm 2 provides a sketch of the overall training process – the $B_w^{\text{EXPLOB}}$ updates are applied in lines 16–17 in the algorithm. We set the hyperparameters $B_{\max}$ and $\lambda$ to be same as $R_{\max}$ ($= 1.0$ for CHAIN).[4]

- $\widehat{R}^{\text{SELFRS}} := \overline{R} + R_\phi^{\text{SELFRS}}$. This corresponds to a part of our reward shaping technique which uses only the intrinsic rewards $R_\phi^{\text{SELFRS}}$. For CHAIN environment, we use a tabular representation for $R_\phi^{\text{SELFRS}}$ where $\phi[s, a]$ reward values are learned for each state-action pair and $R_\phi^{\text{SELFRS}}(s, a) := \phi[s, a] \,\forall (s, a)$. Along with $R_\phi^{\text{SELFRS}}$, a tabular value-function $V_{\overline{R}, \widetilde{\phi}}$ is maintained w.r.t. $\overline{R}$, serving as critic to compute values $V_{\overline{R}}^{\pi_k}(s)$ as needed for the empirical updates (see Section 3.3). For updating $V_{\overline{R}, \widetilde{\phi}}$, we use Monte Carlo updates based on the trajectory returns as target and using a $\ell_2$-norm loss function [7]. Algorithm 2 provides a sketch of the overall training process – the $R_\phi^{\text{SELFRS}}$ updates are applied in lines 11–15 in the algorithm. We set the following values for hyperparameters: (a) learning rate for updating $\phi$ parameters is set to $0.01$; (b) learning rate for updating $\widetilde{\phi}$ parameters is set to $0.01$; (c) frequency of updates $N_r$ is set to be 5, i.e., update after every 5 rollouts in the environment; (d) we have a rollout buffer $\mathcal{D}$ of size 10. Furthermore, in all our experiments with Q-learning agent, we clipped the values of $\phi$ in the range $[-0.01, 0.01]$ (see Section 4.3 and Appendix F for another variant of clipping used with neural agents).

- $\widehat{R}^{\text{EXPLORS}} := \overline{R} + R_\phi^{\text{SELFRS}} + B_w^{\text{EXPLOB}}$. This is our exploration-guided reward shaping technique that combines intrinsic bonuses with intrinsic rewards. Algorithm 2 provides a sketch of the overall training process; we update $R_\phi^{\text{SELFRS}}$ and $B_w^{\text{EXPLOB}}$ in the same way as described in the previous two points above.

Note that, for stability, we update the policy more frequently than the intrinsic reward ($N_\pi = 2$ vs. $N_r = 5$) and at a higher learning rate (0.1 vs. 0.01), as considered in the work of [25, 26]. In the tabular setting with CHAIN, we find that the overall quantitative results are robust to hyperparameters mentioned above – we use the exact same set of hyperparameters for evaluation on ROOM in Section 4.2.

---

[4]In our implementation, we do a more fine-grained update where the counts are updated during the rollout itself, instead of waiting for the end of the rollout. Moreover, in our implementation, the bonus reward given for state-action $(s, a)$ corresponds to bonus associated with the next state $s'$ visited in the rollout.

**Evaluation setup: compute resources.** We ran the experiments on a cluster comprising of machines with 3.30 GHz Intel Xeon CPU E5-2667 v2 processor and 256 GB RAM.

# E Evaluation on ROOM: Additional Details (Section 4.2)

**ROOM (Figure 3).** The environment used in our experiments is based on the work of [54]; however, we adapted it to have a "distractor" state (shown as "green-plus") that could provide a small positive reward. Next, we present additional details about the environment. We represent the environment as an MDP with $\mathcal{S}$ states, each corresponding to cells in the grid-world indicating the agent's current location (shown as "blue-circle"). The goal (shown as "green-star") is located at the top-right corner cell; in the bottom-left room, there can be a "distractor" state (shown as "green-plus") that could provide a small positive reward. The agent can take four actions given by $\mathcal{A} := \{$"up", "left", "down", "right"$\}$. An action takes the agent to the neighbouring cell represented by the direction of the action; however, if there is a wall (shown as "brown-segment"), the agent stays at the current location. There are also a few terminal walls (shown as "thick-red-segment") that terminate the episode, located at the bottom-left corner cell, where "left" and "down" actions terminate the episode. Furthermore, when an agent takes an action $a \in \mathcal{A}$, there is $p_{\text{rand}}$ probability that an action $a' \in \mathcal{A} \setminus \{a\}$ will be executed instead of $a$. The agent gets a reward of $R_{\text{max}}$ after it has navigated to the goal and then takes a "right" action (i.e., the reward can be accumulated in this state); similarly, the "up" action in the distractor state gives a reward of $R_{\text{dis}}$. The reward is 0 for all other state-action pairs. There is a discount factor $\gamma$ and an episode terminates after $H = 30$ steps. The environment-specific parameters (including $p_{\text{rand}}, R_{\text{max}}, R_{\text{dis}}, \gamma$) are kept same as in Section 4.1, i.e., $p_{\text{rand}} = 0.05, R_{\text{max}} = 1, R_{\text{dis}} = 0$ or $0.01$, and $\gamma = 0.99$. Similar to the two variants of CHAIN environment, we have two variants of this environment: (a) ROOM$^0$ has $R_{\text{dis}} = 0$ at the distractor state shown as "green-plus" (equivalently, there is no distractor state); (b) ROOM$^+$ has $R_{\text{dis}} = 0.01$ at the distractor state.

# F Evaluation on LINEK: Additional Details (Section 4.3)

**LINEK (Figure 4).** We expand on the details of the LINEK environment, introduced in Section 4.3. As discussed in Section 4.3, this environment corresponds to a navigation task in a one-dimensional space where the agent has to first pick the correct key and then reach the goal. The environment used in our experiments is based on the work of [54]; however, we adapted it to have multiple keys (only one being correct) and "distractor" states that provide a small reward at goal locations even without the correct key. The environment comprises of the following main elements: (a) an agent whose current location (shown as "blue-circle") is a point x in $[0, 1]$; (b) goal (shown as "green-star") is available in locations on the segment $[0.9, 1]$; (c) a set of $k$ keys that are available in locations on the segment $[0.0, 0.1]$; (d) among $k$ keys, only 1 key is correct and the remaining $k - 1$ keys are wrong (i.e., irrelevant at the goal). The agent's initial location is sampled from $[0.3, 0.4]$.

The agent can take four actions given by $\mathcal{A} := \{$"left", "right", "pickCorrect", "pickWrong"$\}$. "pickCorrect" action does not change the agent's location, however, when executed in locations where keys are available, the agent acquires the correct key required at the goal; if the agent already possesses any key, the action has no effect. "pickWrong" action does not change the agent's location, however, when executed in locations where keys are available, the agent acquires one of the $k - 1$ wrong keys (chosen at random); if agent possesses a key, the action has no effect. A move action of type "left" or "right" takes the agent from the current location in the direction of the move with the dynamics of the final location captured by two hyperparameters $(\Delta_{a,1}, \Delta_{a,2})$; for instance, with current location x and action "left", the new location x' is sampled uniformly among locations from $(x - \Delta_{a,1} - \Delta_{a,2})$ to $(x - \Delta_{a,1} + \Delta_{a,2})$. The agent's move action is not applied if the new location crosses the wall, and there is $p_{\text{rand}}$ probability of a random action.

The agent receives rewards as follows: (a) $R_{\text{max}}$ once it has navigated to the goal location after acquiring the correct key and then takes a "right" action (the action doesn't terminate the episode and reward can be accumulated); (b) $R_{\text{dis}}$ after it has navigated to the goal location without acquiring the correct key and then takes a "right" action (the action doesn't terminate the episode and reward can be accumulated); (c) the reward is 0 elsewhere. We have a discount factor $\gamma$ and the environment resets after a horizon of $H$. We set $p_{\text{rand}} = 0.05, R_{\text{max}} = 1, R_{\text{dis}} = 0$ or $0.01, H = 60, \gamma = 0.99,$ $\Delta_{a,1} = 0.075,$ and $\Delta_{a,2} = 0.01$.

We obtain different variants of the environment by changing the values of $R_{\text{dis}}$ and number of keys $k$. Similar to Sections 4.1 and 4.2, we use two adaptations of the environment: (i) LINEK$^0$ with $(k = 10, R_{\text{dis}} = 0)$ (i.e., without any distractor state); (ii) LINEK$^+$ with $(k = 10, R_{\text{dis}} = 0.01)$ (i.e., with distractor states). In our experiments, we represent the environment as an MDP with $\mathcal{S}$ states comprising of the following: (a) the agent's current location (a point x in $[0, 1]$); (b) one bit indicating if the agent is on a segment with keys; (c) one bit indicating if the agent is on a segment with the goal; (d) $k$ bits, corresponding to each of the $k$ keys, indicating whether agent has that key or not (at most one of these bits can be one, as the agent can acquire only one key at any point in time, according to the transition dynamics specified above). This state representation is the input observation space for neural networks used by our policy and intrinsic reward functions.

**Evaluation setup: agents.** We conduct our experiments with a neural REINFORCE agent using a two-layered neural network architecture (i.e., one fully connected hidden layer with 256 nodes and RELU activation) [7]. In all the experiments that used neural-network based policies for agents, we also kept an exploration factor of $\epsilon = 0.05$, i.e., the agent uses soft-max neural policy with probability $(1 - \epsilon)$ and chooses a random action with $\epsilon$. Algorithm 2 provides a sketch of the overall training process, and shows how agent's training interleaves with reward shaping techniques – the agent's policy is updated in lines 4–8 of the algorithm. For the agent's training process, we use a fixed set of hyperparameters irrespective of the type of reward shaping technique or specific variant of the environment. More concretely, we have the following: (a) the agent's learning rate is set to $10^{-5}$; (b) frequency of updates $N_\pi$ is set to be 2, i.e., update after every 2 rollouts in the environment; (c) a rollout buffer (first-in-first-out) $\mathcal{D}$ of size 10 is maintained and we update the agent's policy using the last 5 rollouts in $\mathcal{D}$. Most of these hyperparameters are close to what we used for the tabular REINFORCE agent in the CHAIN environment, described in Appendix D.

**Evaluation setup: shaping techniques.** Next, we describe different reward shaping techniques used during the agent's training phase. Specifically, during training, the agent receives rewards based on the shaped reward function $\widehat{R}$; the performance (as reported in the plots) is always evaluated w.r.t. the extrinsic reward function $\overline{R}$. Similar to Section 4.1, we compare the performance of six techniques. As a crucial difference, here we use neural-network based reward functions for $\widehat{R}^{\text{SORS'}}$, $\widehat{R}^{\text{LIRPG'}}$, $\widehat{R}^{\text{SELFRS}}$, and $\widehat{R}^{\text{EXPLORS}}$. We provide details of the different reward shaping techniques below:

- $\widehat{R}^{\text{ORIG}} := \overline{R}$. This serves as a default baseline where extrinsic reward function is used during training without any shaping.

- $\widehat{R}^{\text{SORS'}} := \overline{R} + R_\phi^{\text{SORS}}$. This is based on the SORS technique [26]; see additional details in Section 2.2 (also see Footnote 2 about $\widehat{R}^{\text{SORS'}}$). Following the neural architectures used for reward functions in [25, 26], we use the same neural-network architecture as used for the agent's policy – instead of using soft-max at the output layer to compute probability distribution over actions, here we use $tanh$-clipping (with a scaling factor of $0.10$) to get output reward values for actions. Algorithm 2 provides a sketch of the overall training process – the $R_\phi^{\text{SORS}}$ updates would be applied in lines 11–15 in the algorithm. We update the intrinsic reward function using the following hyperparameters: (a) the learning rate is set to $10^{-3}$; (b) frequency of updates $N_r$ is set to be 20, i.e., update after every 20 rollouts in the environment; (c) we have a rollout buffer $\mathcal{D}$ of size 10 and sample a set of 10 *pairs* of rollouts for the gradient updates (in our implementation, we prioritized sampling of pairs that have non-zero gap between returns).

- $\widehat{R}^{\text{LIRPG'}} := \overline{R} + R_\phi^{\text{LIRPG'}}$. This is obtained via adapting the LIRPG technique of [25] to our training pipeline; see Algorithm 2, Sections 2.2 and 3.2 (also see Footnote 3 about $\widehat{R}^{\text{LIRPG'}}$). More specifically, in our experiments, we implemented $\widehat{R}^{\text{LIRPG'}}$ as an adaptation of $\widehat{R}^{\text{SELFRS}}$ where we set $h \to \infty$ instead of 1 in $A_{\widehat{R},h}^{\pi_{\theta_k}}(s, a)$ (see Section 3.2) – the rest of the implementation is same as described below for $\widehat{R}^{\text{SELFRS}}$. When computing $A_{\widehat{R},h}^{\pi_{\theta_k}}(s, a)$ for $h > 1$, we need an additional rollout to be able to compute this quantity. In our experiments with LINEK, we set $h \to \infty$ only for the starting state of the episode and kept $h = 1$ for the rest of the trajectory – this helped in reducing the computation time and variance.

- $\widehat{R}^{\text{EXPLOB}} := \overline{R} + B_w^{\text{EXPLOB}}$. This corresponds to a part of our reward shaping technique which uses only the intrinsic bonuses $B_w^{\text{EXPLOB}}$. As discussed in Sections 3.1 and 3.3, we use a count-based bonus $B_w^{\text{EXPLOB}}$. For this environment, we use an abstraction that discretizes the continuous

location part of the state to 0.1-length segments, i.e., creating 10 segments in total; the bits used to represent different indicator flags are then used along with these segments to represent an abstracted state. Given this abstraction, the rest of the process and hyperparameters for updating $B_w^{\text{EXPLOB}}$ are the same as discussed in Appendix D.

- $\widehat{R}^{\text{SELFRS}} := \overline{R} + R_\phi^{\text{SELFRS}}$. This corresponds to a part of our reward shaping technique which uses only the intrinsic rewards $R_\phi^{\text{SELFRS}}$. By following the neural architectures used for reward functions in [25, 26], we use the same neural-network architecture as used for the agent's policy. In particular, we use two networks for $\widehat{R}^{\text{SELFRS}}$: (a) one network is used for the reward function $R_\phi^{\text{SELFRS}}$ that applies $tanh$-clipping (with a scaling factor of $0.10$) instead of soft-max to get output reward values for actions; (b) the second network is used for learning value-function $V_{\overline{R},\widetilde{\phi}}$ that applies a linear layer instead of a soft-max layer to obtain state-values. For updating $V_{\overline{R},\widetilde{\phi}}$, we use Monte Carlo updates based on the trajectory returns as target and using a $\ell_2$-norm loss function [7]. Algorithm 2 provides a sketch of the overall training process – the $R_\phi^{\text{SELFRS}}$ updates are applied in lines 11–15 in the algorithm. We set the following values for hyperparameters: (a) learning rate for updating $\phi$ parameters is set to $10^{-3}$; (b) learning rate for updating $\widetilde{\phi}$ parameters is set to $5 \cdot 10^{-3}$; (c) frequency of updates $N_r$ is set to be 20, i.e., update after every 20 rollouts in the environment; (d) we have a rollout buffer $\mathcal{D}$ of size 10.

- $\widehat{R}^{\text{EXPLORS}} := \overline{R} + R_\phi^{\text{SELFRS}} + B_w^{\text{EXPLOB}}$. This is our exploration-guided reward shaping technique that combines intrinsic bonuses with intrinsic rewards. Algorithm 2 provides a sketch of the overall training process; we update $R_\phi^{\text{SELFRS}}$ and $B_w^{\text{EXPLOB}}$ in the same way as described in the previous two points above.

We update the policy more frequently than the intrinsic reward ($N_\pi = 2$ vs. $N_r = 20$), as considered in the work of [25, 26]. Moreover, for the first 5000 episodes of training, we do not supply intrinsic reward signals from neural network components of $\widehat{R}^{\text{SORS'}}$, $\widehat{R}^{\text{LIRPG'}}$, $\widehat{R}^{\text{SELFRS}}$, or $\widehat{R}^{\text{EXPLORS}}$ (even though we keep updating their neural network components as usual) – this helps in preventing spuriourous reward signals associated with initialization of neural networks.