# OpenReview forum: "Exploration-Guided Reward Shaping for Reinforcement Learning under Sparse Rewards"
_NeurIPS.cc/2022/Conference — NeurIPS 2022 Accept_

### Official Review · Reviewer_8ukX · 2022-06-29

**Rating:** 6
**Confidence:** 2
**Soundness:** 2 fair
**Presentation:** 3 good
**Contribution:** 2 fair

**Summary:**

The paper takes a holistic approach to reward shaping in RL, and has two main contributions: (1) simplified meta-gradients for updating a parameterized intrinsic reward function, which generalizes beyond policy gradients algorithms, and (2) the combination of said intrinsic reward function with an intrinsic exploration bonus. The approach is self-supervised, and does not require expert knowledge. The combination of different intrinsic signals is studied theoretically in a chain environment, and then evaluated empirically on several low-dimensional MDPs, both in tabular settings and with function approximators.

**Questions:**

**Main comments**
* The title is not very informative. I would suggest to consider something that clearly establishes the existence of two shaping terms, as well as the generalization beyond policy gradient algorithms.
* Similarly, the abstract is also vague. I would recommend to include the concrete contributions of the paper, especially mentioning that the authors generalize self-supervised reward shaping techniques to general RL algorithms. Moreover, the sentence on line 8 does not clarify that two different terms are added to the environment's reward signal.
* Despite generally stating in the abstract that they would 'theoretically showcase the utility of our reward shaping framework', the authors only present a limited theoretical analysis on a particular family of MDPs. Although this is not negative per se, it should be made clear right from the start.
* Is there a reason why LINRPG is not included in the empirical evaluation on REINFORCE? In would be particularly interesting to see how SELFRS compares against it.
* Section 3 could further benefit by clarifying that the derivations until line 163 are reported from existing works. Moreover, it should anticipate that the derivation will skip non-trivial steps, which are however reported in the Appendix.
* On line 190, the paper mentions "distractive" rewards, but I fail to see how the rewards in this particular environment are distractive.

**Minor comments**
* line 20: To the best of my knowledge, transfer learning approaches do not need domain-knowledge, and can rely solely on data. Can this be clarified?
* line 131: The authors should anticipate that superscripts (e.g. SelfRS) are dropped from this point on, as they do in the Appendix.
* line 157: Where is this smoothness condition derived from? is it an assumption?
* line 209: I would suggest rewriting the bound as $2^{n_1} - 1$ for ease of reading.
* possibly due to my lack of familiarity, I would ask the authors to explain why the equality between the last line on page 17, and the first line on page 18 holds.

**Nitpicks**
* line 69: The notation for T does not clarify that its output is a probability distribution over the action space (i.e. integrates to 1).
* line 234: green-star -> blue-star

**Limitations:**

Authors present a sufficient discussion of limitations in their conclusion.

**Strengths And Weaknesses:**

**Strengths**

* The paper is well-written and easy to follow, despite the large quantity of preliminary derivations.
* The integration of both reward shaping terms is an interesting concept, and the analysis in Appendix D sheds nice insights on the synergy of the two.
* Empirical performance appears strong across the environments considered in the paper, and the evaluation protocol seems robust.

**Weaknesses**
* In its main body, the paper does not always clarify which derivations are original, and which are reported from previous works (see following comments).
* A number of separate concerns on the introductory writing and baselines would need to be addressed to further improve the quality of he paper (see following section, main comments).

---

> ### Author Response · Authors · 2022-08-02
> **Response to Reviewer 8ukX**
>
> Thank you for carefully reviewing our paper! We greatly appreciate your feedback. Please see below our responses to your comments.
>
> -----
>
> **1. The title is not very informative... Similarly, the abstract is also vague.**
>
> We thank the reviewer for the feedback, and we will incorporate the suggestions given by the reviewer in the final version of the paper:
> - We will clarify and stress the concrete contributions of the paper in the abstract, e.g., generalizing self-supervised reward shaping techniques to general RL algorithms.
> - We will clarify that two different terms (intrinsic reward and intrinsic bonus) are added to the environment’s reward signal.
>
> -----
> **2. the authors only present a limited theoretical analysis on a particular family of MDPs. Although this is not negative per se, it should be made clear right from the start.**
>
> In the introduction, we will clarify that the current theoretical analysis in Section 3.3 is applicable only for a particular family of deterministic MDPs and a specific learner model. We note that the current analysis could possibly be extended to the following settings:
> - A deterministic MDP represented using a directed graph structure, where the nodes correspond to states and edges correspond to transitions.
> - A more general learner model instead of the simplified update rule currently considered in Algorithm 3.
>
> -----
> **3. Is there a reason why LIRPG is not included in the empirical evaluation on REINFORCE? In would be particularly interesting to see how SelfRS compares against it.**
>
> We thank the reviewer for their feedback about the experimental evaluation. We have now conducted additional experiments and shared the results in a common response to all the reviewers. Please refer to our responses at the top of the page with the following titles:
> - Additional experimental results with Exp and LIRPG: Figures 6a and 6d (Part 1)
> - Additional experimental results with Exp and LIRPG: Figures 6e and 6h (Part 2)
> - Additional experimental results with Exp and LIRPG: Figures 7c and 7d (Part 3)
>
>
> -----
> **4. Section 3 could further benefit by clarifying that the derivations until line 163 are reported from existing works.**
>
> As noted by the reviewer, derivations until line 163 are reported from existing works; the remaining part involves non-trivial steps and we will clarify this point in the final version of the paper.
>
> -----
> **5. On line 190, the paper mentions "distractive" rewards, but I fail to see how the rewards in this particular environment are distractive.**
>
> We thank the reviewer for pointing out this error. We actually meant distractive paths in the state space, and not the distractive rewards. We will fix this error.
>
> -----
> **6. Minor comments**
>
> - **Re. line 20:** Thank you for the suggestion; we will update this remark about transfer learning approaches.
> - **Re. lines 131 and 209:** We will incorporate the reviewer’s suggestions in the final version of the paper.
> - **Re. line 157:** Yes, the smoothness condition is an assumption; we will clearly state this upfront.
> - **Why does the equality between the last line on page 17, and the first line on page 18 holds?**:  It is due to the fact that $1_{s,a}^\top \cdot 1_{s’,a’} = 0$ for $(s,a) \ne (s’,a’)$; we’ll add this intermediate step in the derivation.
>
> -----
>
> We hope that our responses can address your concerns and are helpful in improving your rating. If you have any other comments or feedback, please let us know! We are looking forward to hearing back from you! Thank you again for the review.

---

> > ### Comment · Reviewer_8ukX · 2022-08-05
> > **Reviewer Response**
> >
> > I would like to thank the authors for their clarifications and additional experimental results. I would encourage the authors to directly upload a revision of the paper including the changes discussed in their response, if possible.

---

> ### Author Response · Authors · 2022-08-09
> **Follow up response to Reviewer 8ukX**
>
> Dear Reviewer 8ukX,
>
> We have now uploaded a revised version of the main paper. We have provided a summary of the key changes in a common response to all the reviewers above. We truly believe that the paper has significantly improved based on the feedback, and we really appreciate your input.  We also hope that the revised version is helpful in improving your rating. Please let us know if you have any other comments or feedback. We will be happy to incorporate further feedback in the final revision. We are looking forward to hearing back from you!
>
> Thank you!
>
> Authors

---

### Official Review · Reviewer_tE4G · 2022-07-08

**Rating:** 6
**Confidence:** 4
**Soundness:** 3 good
**Presentation:** 3 good
**Contribution:** 2 fair

**Summary:**

The paper proposes a new method for reward shaping (called EXPRS) that leverages intrinsic rewards from self-supervised reward shaping techniques, as well as exploration bonus in the shaped reward. The authors first describe the general setting of reward shaping followed by existing reward techniques and their shortcomings, specifically self-supervised reward shaping and exploration bonus based reward shaping. Subsequently the authors introduce their reward shaping formulation, which includes both the intrinsic self-supervised reward and the exploration bonus in a combined formulation. The authors then derive gradient update formulas for their proposed algorithm, in a general and empirical setting, and outline the steps in their EXPRS method.

Following the definition of EXPRS, the authors perform a theoretical analysis of how EXPRS should perform in a chain environment compared to other methods and then show experimental results of EXPRS and those methods in a chain environment, as well as a Room environment and a LineK environment. The paper then concludes with a brief discussion of the results and some directions for future work.

**Questions:**

**General Questions:**
- Could you define "b" in Line 167? This is mainly for completeness since I did not see it defined (I might have missed it)
- How general is the gradient derivation to different RL algorithms? For example, if someone wanted to apply EXPRS to PPO, would they have to derive empirical updates just for PPO? It would be good to get a better sense of how easy/difficult it is to plug in EXPRS to different RL algorithms
- Is the theoretical analysis in Section 3.3 mainly limited to the chain environment or does the general utility hold beyond it? I noticed that the appendix includes further details on how you arrived at cost for the different cases, which would be good to mention in the paper.
- What is the reasoning behind using the baselines that you show in your experiments? Also, is LIPRG included there (it could be that SELF-RS is similar to LIPRG but that is currently unclear)?
- It looks like for many of the experiments that you performed, EXPRS and some of the baselines fall within the deviation of each other. Could you comment on where you see clear advantages of EXPRS and where it is more difficult to distinguish its performance from other methods.

**Limitations:**

I think the paper could be improved by a more thorough discussion of the limitations of EXPRS, which are only briefly touched upon in the conclusion. It would be good to get more detail on what the authors anticipate would be needed (in general terms) to perform the extensions that they mention (continuous spaces, converge, other agents). Given space limitations, it would be OK to include this in the appendix even though I would recommend it to be in the main paper.

**Strengths And Weaknesses:**

**Originality:**
- Strengths: The paper proposes a new method for automated for reward shaping in RL that draws upon two prior methods and combines them into a single formulation. Relevant work is cited and compared against in the beginning of the paper.
- Weaknesses: The authors could have provided more detail on the limitations of their method, which are only briefly discussed in the conclusion.

**Quality:**
- Strengths: The paper provides a good amount of detail in the derivation steps related to the method and is generally very thorough in description of relevant details.
- Weaknesses: The number of experiments that showcase the capabilities of the method is somewhat limited.

**Clarity:**
- Strengths: The paper is generally well organized and includes relevant figures and diagrams for the environments used in the experiments.
- Weaknesses: The authors could have helped guide readers by providing summaries of important points/messages throughout the paper, especially after sections with heavy text and equations. For example, after Section 2 it could be helpful to have a small table outlining the prior methods and outline what they can do (e.g. with checkmarks) and how EXPRS has capabilities beyond those method. In Section 3.2 it could be helpful to state in the beginning what the missing terms in the gradient computation are and how the derivation leads to them being available when describing the final algorithm.

**Significance:**
- Strengths: The paper proposes a new method for exploration based in reward shaping, which is a relevant problem in RL.
- Weaknesses: The environments and algorithmic varieties for which the method was tested were somewhat limited and provide little insight into how the method may perform in more challenging settings.

---

> ### Author Response · Authors · 2022-08-02
> **Response to Reviewer tE4G**
>
> Thank you for carefully reviewing our paper! We greatly appreciate your feedback. Please see below our responses to your comments.
>
> -----
> **1. How general is the gradient derivation to different RL algorithms?**
>
> The final form of the empirical update rule for ExpRS (in particular, see Eq. 2) does not rely on specifics of any particular RL algorithm; infact, the update rule in Eq. 2 is compatible with any RL algorithm and not just policy-gradient style methods. For instance, in the paper, we demonstrated the effectiveness of ExpRS for Q-learning agent. We note that SORS is also compatible with any RL algorithm; in contrast, LIRPG is only applicable to policy-gradient style methods.
>
> -----
> **2. Is the theoretical analysis in Section 3.3 mainly limited to the chain environment...?**
>
> The current theoretical analysis in Section 3.3 is applicable to a particular family of deterministic MDPs and a specific learner model. We note that the current analysis could possibly be extended to the following settings:
> - A deterministic MDP represented using a directed graph structure, where the nodes correspond to states and edges correspond to transitions.
> - A more general learner model instead of the simplified update rule currently considered in Algorithm 3.
>
> -----
> **3. What is the reasoning behind using the baselines that you show in your experiments? Also, is LIRPG included there..?**
>
> In the supplementary material (neurips2022_exprs_supp.pdf), we have provided a more detailed discussion of the techniques used in our experimental evaluation. Please refer to the following:
> - Appendix E, lines 749-785
> - Appendix F, lines 826-828
> - Appendix G, lines 893-930
>
> Moreover, we have conducted additional experiments and shared the results in a common response to all the reviewers. Please refer to our responses at the top of the page with the following titles:
> - Additional experimental results with Exp and LIRPG: Figures 6a and 6d (Part 1)
> - Additional experimental results with Exp and LIRPG: Figures 6e and 6h (Part 2)
> - Additional experimental results with Exp and LIRPG: Figures 7c and 7d (Part 3)
>
> -----
> **4. …comment on where you see clear advantages of ExpRS and where it is more difficult to distinguish its performance from other methods.**
>
> In the supplementary material (neurips2022_exprs_supp.pdf), we have done a systematic empirical investigation in Appendices E and G. We kindly request the reviewer to check the experimental results reported in Figures 6 and 7. In particular, we have summarized the main experimental results in the following:
> - Appendix E, lines 796-804
> - Appendix F, lines 826-832
> - Appendix G, lines 939-950
>
> In short, we expect ExpRS to have a substantial benefit over baselines (SelfRS, Exp, SORS, LIRPG, Orig) in domains that have some of the following characteristics: (i) sparse rewards (hard exploration), (ii) distractive states that may serve as sub-optimal goals (local minima), (iii) exploratory pathways without any rewards (noisy TV problem). Our experimental results in Figures 6 and 7 showcase that ExpRS is the most robust and best-performing technique across all the experimental settings.
>
> -----
> **5. …the paper could be improved by a more thorough discussion of the limitations of ExpRS…**
>
> Below we share some thoughts on the limitations of ExpRS and potential extensions.
> - First, we note that the ExpRS technique could possibly hurt the learning progress of an agent; in fact, this is a limitation of any self-supervised reward shaping technique. It would be interesting to theoretically investigate the robustness of ExpRS and develop a deeper understanding of when we can expect a guaranteed performance.
> - Second, the experimental evaluation is focused on simpler environments to study the performance of techniques w.r.t. the three characteristics of (a) hard exploration, (b) local minima, and (c) noisy TV problem. It would be interesting to evaluate different reward design techniques in more complex environments and compare the performance of ExpRS, SORS, and LIRPG. This would also require designing benchmark environments that systematically capture the above three characteristics.
> - Third, ExpRS combines the intrinsic rewards and intrinsic bonuses that allows it to overcome the limitations of state-of-the-art techniques. It would be interesting to investigate this idea further and develop more principled ways to combine these two signals.
>
> -----
> **6. Additional comments**
>
> - **Could you define "b" in Line 167?**: It is a summation over the action set, i.e., $b \in \mathcal{A}$; we will clarify this in the paper.
> - **Suggestions about clarity**: Thank you for the feedback. In the final draft, we will incorporate the reviewer’s suggestions, e.g., guide the readers by providing summaries of important points in relevant places.
>
> -----
>
> We hope that our responses can address your concerns and are helpful in improving your rating. If you have any other comments or feedback, please let us know!

---

> > ### Comment · Reviewer_tE4G · 2022-08-06
> > **Reply to Author Response**
> >
> > I appreciate the authors' thorough response to my review, questions and feedback. Given the response, I have adjusted my score. Some questions and comments that remain on my mind:
> > - The new results you in the Appendix show the ExpRS and SELF-RS generally perform very similar across almost all experiments. Could you comment more on what you view as the distinction between the two (other than the missing exploration bonus term) and in which situations they could be applicable?
> > - Given the new set of results and extensive appendix, it has gotten easier to get lost in a lot of data. I think that it would benefit the authors to restructure some of their paper to showcase which results they would like readers to focus on.

---

> ### Author Response · Authors · 2022-08-09
> **Follow up response to Reviewer tE4G**
>
> Dear Reviewer tE4G,
>
> We have now uploaded a revised version of the main paper. We have provided a summary of the key changes in a common response to all the reviewers above.
>
> ----
> In the revised paper, the experimental results in Figure 2 and Figure 4 more clearly demonstrate the effectiveness and robustness of ExpRS across different environments in comparison to all the baselines (Orig, SORS, LIRPG, Exp, SelfRS). We want to highlight two specific points:
>
> - Results in Figure 2d, 4b, 4d show that SelfRS and SORS could completely fail in agent’s training when the environment has a state with distractive rewards.
>
> - Results in Figure 4d for $\text{LineKey}^+$ environment show the brittleness of baselines techniques (SORS, LIRPG, SelfRS) in sparse/noisy environments – here, these techniques lead to a performance even worse compared to that of Orig.
>
> ----
>
> We truly believe that the paper has significantly improved based on the feedback, and we really appreciate your input. We also hope that the revised version is helpful in improving your rating. Please let us know if you have any other comments or feedback. We will be happy to incorporate further feedback in the final revision. We are looking forward to hearing back from you!
>
> Thank you!
>
> Authors

---

### Official Review · Reviewer_JLEi · 2022-07-11

**Rating:** 4
**Confidence:** 3
**Soundness:** 3 good
**Presentation:** 3 good
**Contribution:** 3 good

**Summary:**

The paper studies the problem of learning informative rewards, without domain knowledge or external supervision, that will accelerate learning in extremely sparse reward settings with noisy distractions. To address this problem, the paper proposes Exploration-Guided Reward Shaping (ExpRS) that balances exploration and explotation of environment rewards. The paper derives the update rules for ExpRS in a general setting, not specific to policy gradients. The paper then shows results on three environments comparing to other reward shaping methods.

**Questions:**

* As the authors state, [1] compares on the same set of environments. However, [1] uses a exponential scale on the x-axis. This is important because it shows that the method from [1] takes exponentially less experience to learn. Why did this work not use an exponential x-axis scale?
* In Theorem 1, it seems using an exploration reward contributes more to the cost than the self-supervised reward shaping. Why is this the case?
* Given LIPRG [2] was mentioned several times throughout the paper, why not compare to LIPRG as well in the REINFORCE experiments?
* Why not compare to ExpRD from [1] given that work shows results on the same set of environments?

* The derivations in Eq. 1 detract from the description of the algorithm. I recommend moving the derivation to the supplementary and stating the final result in the main paper.
* The supplementary material also includes the main paper, reformat to only include supplementary material or move the supplementary to the main paper.

[1] Devidze Rati et al, Explicable reward design for reinforcement learning agents. In NeurIPS, 2021.

[2] Zheng Zeyu et al. On learning intrinsic rewards for policy gradient methods. In NeurIPS, 2018.

**Limitations:**

The paper acknowledges that it would be interesting to extend the evaluation to more complex tasks. The paper also discusses the limitations of not having a rigerous theoretical analysis of the method in terms of convergence speed and stability. Despite this discussion, the paper does not sufficiently analyze limitations of the method. In what settings would ExpRD not perform as well? What are the limitations of learning a more complex reward in ExpRD?


**Strengths And Weaknesses:**

Strengths:
* The proposed method is intuitive and to the best of my knowledge novel. The paper discusses benefits of both an intrinsic shaped reward and intrinsic bonus rewards and describes how to combine both into one algorithm. The paper also derives an update rule that works for general RL algorithms, not just policy gradient methods.
* Theorem 1 with the analyzed chain environment is helpful for gaining insight into the utility of ExpRS.
* ExpRS is more robust to distractors in the environment than baselines. The results demonstrate that distractors hurt the performance of all baselines, whereas ExpRS is more robust and affected less.

Weaknesses:
* The simple environments are not enough to verify the algorithm. While these environments are still helpful, results on more complex environments are important to verify the utility of ExpRS. For example, LIPRG [1] a work cited throughout this  paper and published at NeurIPS 2018, shows results on pixel-based Atari games. My concerns are exacerbated with the performance on the LineK task, which is the only task that demonstrates a neural network policy, yet the gap between ExpRS and baselines is small. In LineK, ExpRS is only slightly more sample efficient than the regular sparse reward function. This concerns me that ExpRS cannot scale to neural policies and rewards in complex environments.
* How does ExpRS balance the SelfRS and count-based bonus reward? Since the main contribution of the paper is combining these two rewards, studying their balance is importance. On line 183, the paper shows the count-based reward is weighted by parameter $\lambda$. The authors should analyze the impact of different reward weightings to see the impact on performance.
* The SORS baseline is not explained enough in Section 4.1. SORS is briefly explained on L114, but a more detailed description would be useful in Section 4 for how the SORS reward is obtained and how it compares to ExpRS.
* The paper might also be lacking some important baselines, which I discuss in the Questions section.

[1] Zheng Zeyu et al. On learning intrinsic rewards for policy gradient methods. In NeurIPS, 2018.

---

> ### Author Response · Authors · 2022-08-02
> **Response to Reviewer JLEi (Part 1)**
>
> Thank you for carefully reviewing our paper! We greatly appreciate your feedback. Please see below our responses to your comments.
>
> -----
> **1. … the ExpRD method from [Rati et al. 2021] takes exponentially less experience to learn. Why did this work not use an exponential x-axis scale? Why not compare to ExpRD from [Rati et al. 2021]...?**
>
> ExpRD from [Rati et al. 2021] is a teacher-focused reward design technique that has access to a teacher policy. In contrast, our ExpRS method is a self-supervised reward design technique. The objectives of ExpRD and ExpRS are different. In ExpRD, given the domain knowledge in the form of a teacher policy, the objective is to design an informative and interpretable reward function. In contrast, in ExpRS, the objective is to perform online reward shaping in a self-supervised manner such that the learning progress of the RL agent is accelerated. Thus, the results in our work are not comparable to the results in [Rati et al. 2021].
>
> -----
> **2. In Theorem 1, it seems using an exploration reward contributes more to the cost than the self-supervised reward shaping. Why is this the case?**
>
> In the sparse reward goal-oriented setting (e.g., the chain environment considered in Theorem 1), the intrinsic reward component will not get updated until the agent obtains the first successful rollout. Once the first successful rollout is obtained, the intrinsic reward component quickly exploits this information. However, to obtain the first successful rollout, the RL agent relies on guided exploration over state space using the exploration bonus. Thus, the exploration bonus term dominates the cost of learning the optimal behavior. Please see Appendix D for the complete proof of Theorem 1.
>
> -----
> **3. Given LIRPG was mentioned several times throughout the paper, why not compare to LIRPG as well in the REINFORCE experiments?**
>
> We thank the reviewer for their feedback about the experimental evaluation. We have now conducted additional experiments and shared the results in a common response to all the reviewers. Please refer to our responses at the top of the page with the following titles:
> - Additional experimental results with Exp and LIRPG: Figures 6a and 6d (Part 1)
> - Additional experimental results with Exp and LIRPG: Figures 6e and 6h (Part 2)
> - Additional experimental results with Exp and LIRPG: Figures 7c and 7d (Part 3)
>
> -----
> **4. … In LineK, ExpRS is only slightly more sample efficient than the regular sparse reward function … ExpRS cannot scale to neural policies and rewards in complex environments.**
>
> In the supplementary material (neurips2022_exprs_supp.pdf), we have done a systematic empirical investigation in Appendices E and G. We kindly request the reviewer to check the experimental results reported in Figures 6 and 7. In particular, we have summarized the main experimental results in the following:
> - Appendix E, lines 796-804
> - Appendix F, lines 826-832
> - Appendix G, lines 939-950
>
> In short, we expect ExpRS to have a substantial benefit over baselines (SelfRS, Exp, SORS, LIRPG, Orig) in domains that have some of the following characteristics: (i) sparse rewards (hard exploration), (ii) distractive states that may serve as sub-optimal goals (local minima), (iii) exploratory pathways without any rewards (noisy TV problem). Our experimental results in Figures 6 and 7 showcase that ExpRS is the most robust and best-performing technique across all the experimental settings.
>
> -----
>
> **5. How does ExpRS balance the SelfRS and count-based bonus reward? … On line 183, the count-based reward is weighted by parameter $\lambda$.**
>
> In our experiments, we set the parameter $\lambda=1$ and didn’t tune this value in any of the experiments. This default choice essentially makes the count-based bonus rewards start from the value $\overline{R}_\text{max}$ and decay afterward. However, it would indeed be interesting to investigate further how to appropriately scale and combine these two signals for a particular environment.
>
> In terms of our theoretical analysis, we only need to set a very small value for $\lambda$ compared to the maximum extrinsic reward, i.e., $\lambda \ll \overline{R}_\text{max}$.   As illustrated in the theoretical analysis (Theorem 1) of a particular family of MDPs with both sparse reward and non-rewarding paths, the SelfRS component quickly exploits the extrinsic reward signal $\overline{R}_\text{max}$ once the first successful rollout is obtained. Hence, the exploration bonus term mainly plays a role until the first successful rollout is obtained. In the analysis, we used $\lambda < \gamma^{n_1} \cdot \overline{R}_\text{max}$. Please see the theoretical analysis in Appendix D. We will add more discussion on the choice of $\lambda$ in the final version of the paper.
>
> -----
> (the response is continued in Part 2)

---

> ### Author Response · Authors · 2022-08-02
> **Response to Reviewer JLEi (Part 2)**
>
> (continuation of the response from Part 1)
>
> -----
> **6. The SORS baseline is not explained enough … a more detailed description would be useful in Section 4 for how the SORS reward is obtained and how it compares to ExpRS.**
>
> Thank you for the feedback. Currently, our supplementary material (neurips2022_exprs_supp.pdf) provides some additional details about SORS in Appendix B.1, lines 539-555. As suggested by the reviewer, we will provide a detailed description of SORS in Section 4.
>
> -----
> **7. … despite this discussion, the paper does not sufficiently analyze limitations of the method. In what settings would ExpRS not perform as well? …**
>
> Below we share some thoughts on the limitations of ExpRS and potential extensions.
> - First, we note that the ExpRS technique could possibly hurt the learning progress of an agent; in fact, this is a limitation of any self-supervised reward shaping technique. It would be interesting to theoretically investigate the robustness of ExpRS and develop a deeper understanding of when we can expect a guaranteed performance.
> - Second, the experimental evaluation is focused on simpler environments to study the performance of techniques w.r.t. the three characteristics of (a) hard exploration, (b) local minima, and (c) noisy TV problem. It would be interesting to evaluate different reward design techniques in more complex environments and compare the performance of ExpRS, SORS, and LIRPG. This would also require designing benchmark environments that systematically capture the above three characteristics.
> - Third, ExpRS combines the intrinsic rewards and intrinsic bonuses that allows it to overcome the limitations of state-of-the-art techniques. It would be interesting to investigate this idea further and develop more principled ways to combine these two signals.
>
> -----
> **8. Additional comments**
>
> In the final version of the paper, we will incorporate the suggestions provided by the reviewer to improve the readability, including:
> - moving the derivation of Eq. 1 to the supplementary and stating the final result in the main paper.
> - reformatting and moving part of the supplementary to the main paper.
>
>
> -----
>
> We hope that our responses can address your concerns and are helpful in improving your rating. If you have any other comments or feedback, please let us know! We are looking forward to hearing back from you! Thank you again for the review.

---

> ### Author Response · Authors · 2022-08-09
> **Follow up response to Reviewer JLEi**
>
> Dear Reviewer JLEi,
>
> We have now uploaded a revised version of the main paper. We have provided a summary of the key changes in a common response to all the reviewers above.
>
> In Section 4 on Experimental Evaluation, we have incorporated two major updates based on the reviewers' suggestions. First, we have added results for two new baseline techniques (Exp, LIRPG) to all the experiments as per our responses below. Second, we have now moved important environment settings from appendices to the main paper that highlight the three characteristics of (a) hard exploration, (b) local minima, and (c) noisy TV problem.
>
> In the revised paper, the experimental results in Figure 2 and Figure 4 more clearly demonstrate the effectiveness and robustness of ExpRS across different environments in comparison to all the baselines (Orig, SORS, LIRPG, Exp, SelfRS).
>
> We truly believe that the paper has significantly improved based on the feedback, and we really appreciate your input. We also hope that the revised version is helpful in improving your rating. Please let us know if you have any other comments or feedback. We will be happy to incorporate further feedback in the final revision. We are looking forward to hearing back from you!
>
> Thank you!
>
> Authors

---

### Author Response · Authors · 2022-08-02
**Additional experimental results with Exp and LIRPG: Figures 6a and 6d (Part 1)**

We thank the reviewers for their feedback about the experimental evaluation. In this common response to all the reviewers, we would like to share additional experimental results with two techniques:
- Exp: Reward shaping technique which uses only the intrinsic bonuses. The experimental results for this technique were already part of the supplementary material (Appendices E, G).
- LIRPG: Reward shaping technique based on LIRPG, adapted to our implementation pipeline. This is a generalization of our reward shaping technique SelfRS where we set $h \to \infty$ instead of $1$ in $\nabla_\phi A^{\pi_k}_{\widehat{R}, h}(s_i, a_i)$. We conducted additional experiments with this technique during the rebuttal phase based on the reviewers’ feedback.

This response is split across three parts, and we are reporting results in the environments listed below.
- Part 1 reports results corresponding to Figures 6a and 6d in the supplementary material (Appendix E) with the LIRPG technique added. In short, these figures are for $\text{Chain}^0_{n_2}$ environment without any distractor state for $n_2=20$ and $n_2=40$. Please refer to Appendix E for details.
- Part 2 reports results corresponding to Figures 6e and 6h in the supplementary material (Appendix E) with the LIRPG technique added. In short, these figures are for $\text{Chain}^+_{n_2}$ environment with a distractor state for $n_2=20$ and $n_2=40$. Please refer to Appendix E for details.
- Part 3 reports results corresponding to Figures 7c and 7d in the supplementary material (Appendix G) with the LIRPG technique added. In short, these figures are for $\text{LineKmc}^+_{k}$ environment with a distractor state for $k=2$ and $k=10$. Please refer to Appendix G for details.

Based on these initial results, we observe that ExpRS is the most robust and best-performing technique across all the experimental settings. LIRPG, similar to SORS and SelfRS, can perform suboptimally when the environment has sparse rewards or distractive states.

**Figure 6a with additional techniques: $\text{Chain}^0_{n_2=20}$**

| Method | 0 | 10K | 20K | 30K | 40K | 50K | 60K | 70K | 80K | 90K | 100K |
| ----- | --- | --- | --- | --- | --- | --- | --- | --- | --- | --- | --- |
| ExpRS | $0.00 \pm 0.0$ | $5.97 \pm 0.9$ | $8.95 \pm 0.4$ | $8.97 \pm 0.4$ | $9.04 \pm 0.4$ | $8.79 \pm 0.4$ | $8.84 \pm 0.4$ | $9.12 \pm 0.4$ |$8.78 \pm 0.4$ | $8.94 \pm 0.4$ | $8.90 \pm 0.4$ |
| SelfRS | $0.00 \pm 0.0$ | $5.49 \pm 1.0$ | $5.79 \pm 1.0$ | $7.04 \pm 0.9$ | $7.58 \pm 0.7$ | $8.45 \pm 0.6$ | $8.62 \pm 0.6$ | $8.60 \pm 0.5$ | $9.08 \pm 0.4$ | $9.01 \pm 0.4$ | $8.98 \pm 0.4$ |
| Exp | $0.00 \pm 0.0$ | $4.80 \pm 0.9$ | $5.43 \pm 1.0$ | $5.93 \pm 0.9$ | $8.36 \pm 0.4$ | $8.83 \pm 0.4$ | $8.64 \pm 0.4$ | $9.13 \pm 0.4$ | $8.86 \pm 0.4$ | $8.94 \pm 0.4$ | $8.99 \pm 0.4$ |
| SORS | $0.00 \pm 0.0$ | $5.16 \pm 1.0$ | $5.68 \pm 1.0$ | $6.63 \pm 0.9$ | $7.34 \pm 0.8$ | $8.32 \pm 0.6$ | $8.35 \pm 0.5$ | $8.55 \pm 0.5$ | $8.85 \pm 0.5$ | $8.63 \pm 0.5$ | $8.64 \pm 0.5$ |
| Orig | $0.00 \pm 0.0$ | $0.02 \pm 0.0$ | $4.18 \pm 0.9$ | $5.24 \pm 1.0$ | $5.48 \pm 1.0$ | $5.47 \pm 1.0$ | $5.22 \pm 1.0$ | $5.50 \pm 1.0$ | $5.65 \pm 1.0$ | $5.59 \pm 1.0$ | $5.53 \pm 1.0$ |
| LIRPG | $0.00 \pm 0.0$ | $5.15 \pm 1.0$ | $5.65 \pm 1.0$ | $6.08 \pm 0.9$ | $6.64 \pm 0.9$ | $7.02 \pm 0.8$ | $7.00 \pm 0.8$ | $7.88 \pm 0.7$ | $7.64 \pm 0.7$ | $7.80 \pm 0.7$ | $7.97 \pm 0.7$ |

**Figure 6d with additional techniques: $\text{Chain}^0_{n_2=40} \text{(Zoomed)}$**

| Method | 0 | 3K | 6K | 9K | 12K | 15K | 18K | 21K | 24K | 27K | 30K |
| ----- | --- | --- | --- | --- | --- | --- | --- | --- | --- | --- | --- |
| ExpRS | $0.00 \pm 0.0$ | $0.04 \pm 0.0$ | $13.00 \pm 0.3$ | $13.53 \pm 0.1$ | $13.61 \pm 0.1$ | $13.57 \pm 0.1$ | $13.65 \pm 0.1$ | $13.58 \pm 0.1$ |$13.46 \pm 0.1$ | $13.82 \pm 0.1$ | $13.78 \pm 0.1$ |
| SelfRS | $0.01 \pm 0.0$ | $0.00 \pm 0.0$ | $01.00 \pm 0.5$ | $06.61 \pm 1.2$ | $11.51 \pm 0.8$ | $13.02 \pm 0.5$ | $13.26 \pm 0.5$ | $13.21 \pm 0.5$ | $13.34 \pm 0.5$ | $13.22 \pm 0.5$ | $13.21 \pm 0.5$ |
| Exp | $0.00 \pm 0.0$ | $0.00 \pm 0.0$ | $00.03 \pm 0.0$ | $00.04 \pm 0.0$ | $02.60 \pm 0.5$ | $10.67 \pm 0.3$ | $12.64 \pm 0.1$ | $12.70 \pm 0.1$ | $13.09 \pm 0.1$ | $13.27 \pm 0.1$ | $12.87 \pm 0.2$ |
| SORS | $0.00 \pm 0.0$ | $0.00 \pm 0.0$ | $00.12 \pm 0.0$ | $05.27 \pm 0.7$ | $10.50 \pm 0.7$ | $11.90 \pm 0.6$ | $13.22 \pm 0.2$ | $13.35 \pm 0.1$ | $13.49 \pm 0.1$ | $13.56 \pm 0.1$ | $13.65 \pm 0.1$ |
| Orig | $0.00 \pm 0.0$ | $0.00 \pm 0.0$ | $00.00 \pm 0.0$ | $00.00 \pm 0.0$ | $00.00 \pm 0.0$ | $00.00 \pm 0.0$ | $00.00 \pm 0.0$ | $00.00 \pm 0.0$ | $00.00 \pm 0.0$ | $00.00 \pm 0.0$ | $00.00 \pm 0.0$ |
| LIRPG | $0.00 \pm 0.0$ | $0.00 \pm 0.0$ | $00.00 \pm 0.0$ | $01.84 \pm 1.2$ | $03.76 \pm 1.6$ | $06.38 \pm 1.8$ | $07.98 \pm 1.7$ | $08.14 \pm 1.7$ | $07.36 \pm 1.7$ | $08.20 \pm 1.7$ | $08.47 \pm 1.8$ |


-----
(the response is continued in Part 2)

---

### Author Response · Authors · 2022-08-02
**Additional experimental results with Exp and LIRPG: Figures 6e and 6h (Part 2)**

(continuation of the response from Part 1)

-----

**Figure 6e with additional techniques: $\text{Chain}^+_{n_2=20}$**

| Method | 0 | 10K | 20K | 30K | 40K | 50K | 60K | 70K | 80K | 90K | 100K |
| ----- | --- | --- | --- | --- | --- | --- | --- | --- | --- | --- | --- |
| ExpRS | $0.00 \pm 0.0$ | $5.64 \pm 1.0$ | $8.82 \pm 0.5$ | $8.66 \pm 0.5$ | $8.85 \pm 0.5$ | $8.61 \pm 0.5$ | $8.71 \pm 0.5$ | $8.77 \pm 0.5$ |$8.72 \pm 0.5$ | $8.82 \pm 0.5$ | $8.67 \pm 0.5$ |
| SelfRS | $0.00 \pm 0.0$ | $5.49 \pm 1.0$ | $5.50 \pm 1.0$ | $7.08 \pm 0.8$ | $7.83 \pm 0.7$ | $7.72 \pm 0.7$ | $7.78 \pm 0.7$ | $8.10 \pm 0.7$ | $7.99 \pm 0.7$ | $7.94 \pm 0.7$ | $7.97 \pm 0.7$ |
| Exp | $0.00 \pm 0.0$ | $5.02 \pm 0.9$ | $5.37 \pm 1.0$ | $5.85 \pm 1.0$ | $8.36 \pm 0.5$ | $8.63 \pm 0.5$ | $8.61 \pm 0.4$ | $8.79 \pm 0.4$ | $8.92 \pm 0.4$ | $8.84 \pm 0.4$ | $8.97 \pm 0.4$ |
| SORS | $0.00 \pm 0.0$ | $2.28 \pm 0.8$ | $2.63 \pm 0.9$ | $2.65 \pm 0.9$ | $2.58 \pm 0.8$ | $2.52 \pm 0.8$ | $2.70 \pm 0.9$ | $2.65 \pm 0.9$ | $2.67 \pm 0.9$ | $2.68 \pm 0.9$ | $2.67 \pm 0.9$ |
| Orig | $0.00 \pm 0.0$ | $0.00 \pm 0.0$ | $3.85 \pm 0.9$ | $5.41 \pm 1.0$ | $5.37 \pm 1.0$ | $5.39 \pm 1.0$ | $5.53 \pm 1.0$ | $5.47 \pm 1.0$ | $5.47 \pm 1.0$ | $5.70 \pm 1.0$ | $5.47 \pm 1.0$ |
| LIRPG | $0.00 \pm 0.0$ | $5.59 \pm 1.0$ | $5.50 \pm 1.0$ | $5.49 \pm 1.0$ | $5.57 \pm 1.0$ | $5.49 \pm 1.0$ | $5.71 \pm 1.0$ | $6.65 \pm 0.9$ | $6.83 \pm 0.9$ | $6.94 \pm 0.9$ | $6.67 \pm 0.9$ |

**Figure 6h with additional techniques: $\text{Chain}^+_{n_2=40} \text{(Zoomed)}$**

| Method | 0 | 3K | 6K | 9K | 12K | 15K | 18K | 21K | 24K | 27K | 30K |
| ----- | --- | --- | --- | --- | --- | --- | --- | --- | --- | --- | --- |
| ExpRS | $0.00 \pm 0.0$ | $0.02 \pm 0.0$ | $11.51 \pm 0.7$ | $13.66 \pm 0.1$ | $13.70 \pm 0.1$ | $13.77 \pm 0.1$ | $13.50 \pm 0.2$ | $13.67 \pm 0.1$ |$13.83 \pm 0.1$ | $13.64 \pm 0.1$ | $13.76 \pm 0.1$ |
| SelfRS | $0.00 \pm 0.0$ | $0.00 \pm 0.0$ | $00.90 \pm 0.5$ | $06.73 \pm 1.2$ | $13.15 \pm 0.4$ | $13.77 \pm 0.1$ | $13.48 \pm 0.1$ | $13.37 \pm 0.2$ | $13.58 \pm 0.1$ | $13.57 \pm 0.1$ | $13.70 \pm 0.1$ |
| Exp | $0.00 \pm 0.0$ | $0.00 \pm 0.0$ | $00.00 \pm 0.0$ | $00.14 \pm 0.1$ | $02.97 \pm 0.7$ | $11.54 \pm 0.3$ | $12.56 \pm 0.2$ | $12.87 \pm 0.2$ | $13.18 \pm 0.1$ | $13.39 \pm 0.1$ | $13.45 \pm 0.1$ |
| SORS | $0.00 \pm 0.0$ | $0.02 \pm 0.0$ | $00.08 \pm 0.0$ | $00.09 \pm 0.0$ | $00.09 \pm 0.0$ | $00.09 \pm 0.0$ | $00.09 \pm 0.0$ | $00.09 \pm 0.0$ | $00.09 \pm 0.0$ | $00.09 \pm 0.0$ | $00.09 \pm 0.0$ |
| Orig | $0.00 \pm 0.0$ | $0.00 \pm 0.0$ | $00.00 \pm 0.0$ | $00.00 \pm 0.0$ | $00.00 \pm 0.0$ | $00.00 \pm 0.0$ | $00.00 \pm 0.0$ | $00.00 \pm 0.0$ | $00.00 \pm 0.0$ | $00.00 \pm 0.0$ | $00.00 \pm 0.0$ |
| LIRPG | $0.00 \pm 0.0$ | $0.00 \pm 0.0$ | $00.00 \pm 0.0$ | $00.85 \pm 0.8$ |$03.81 \pm 1.6$ | $07.00 \pm 1.7$ | $08.15 \pm 1.7$ | $08.33 \pm 1.8$ | $08.00 \pm 1.7$ | $08.26 \pm 1.7$ | $08.20 \pm 1.7$ |



-----
(the response is continued in Part 3)

---

### Author Response · Authors · 2022-08-02
**Additional experimental results with Exp and LIRPG: Figures 7c and 7d (Part 3)**

(continuation of the response from Part 2)

-----

**Figure 7c with additional techniques: $\text{LineKmc}^+_{k=2}$**

| Method | 0 | 10K | 20K | 30K | 40K | 50K |
| ----- | --- | --- | --- | --- | --- | --- |
| ExpRS | $0.00 \pm 0.0$ | $9.57 \pm 0.9$ | $12.54 \pm 0.7$ | $13.43 \pm 0.8$ | $13.12 \pm 0.7$ | $13.95 \pm 0.8$ |
| SelfRS | $0.00 \pm 0.0$ | $1.38 \pm 0.7$ | $02.36 \pm 1.0$ | $03.28 \pm 1.2$ | $03.66 \pm 1.3$ | $04.01 \pm 1.4$ |
| Exp | $0.00 \pm 0.0$ | $7.84 \pm 0.7$ | $11.96 \pm 0.5$ | $12.83 \pm 0.4$ | $13.77 \pm 0.3$ | $13.31 \pm 0.4$ |
| SORS | $0.00 \pm 0.0$ | $0.81 \pm 0.4$ | $02.68 \pm 1.1$ | $02.81 \pm 1.1$ | $03.38 \pm 1.4$ | $03.08 \pm 1.3$ |
| Orig | $0.00 \pm 0.0$ | $2.11 \pm 0.7$ | $08.35 \pm 1.0$ | $12.39 \pm 0.6$ | $13.82 \pm 0.5$ | $13.34 \pm 0.5$ |
| LIRPG | $0.00 \pm 0.0$ | $1.49 \pm 0.5$ | $07.22 \pm 1.2$ | $10.08 \pm 0.9$ | $10.99 \pm 0.9$ | $12.40 \pm 0.8$ |

**Figure 7d with additional techniques: $\text{LineKmc}^+_{k=10}$**

| Method | 0 | 10K | 20K | 30K | 40K | 50K |
| ----- | --- | --- | --- | --- | --- | --- |
| ExpRS | $0.00 \pm 0.0$ | $6.23 \pm 0.6$ | $13.16 \pm 0.6$ | $14.20 \pm 0.4$ | $14.19 \pm 0.4$ | $13.95 \pm 0.4$ |
| SelfRS | $0.00 \pm 0.0$ | $0.25 \pm 0.0$ | $00.38 \pm 0.1$ | $00.65 \pm 0.3$ | $00.26 \pm 0.0$ | $00.26 \pm 0.0$ |
| Exp | $0.00 \pm 0.0$ | $2.28 \pm 0.4$ | $11.60 \pm 0.5$ | $13.43 \pm 0.5$ | $12.77 \pm 0.5$ | $13.63 \pm 0.4$ |
| SORS | $0.00 \pm 0.0$ | $0.26 \pm 0.0$ | $00.28 \pm 0.0$ | $00.27 \pm 0.0$ | $00.28 \pm 0.0$ | $00.28 \pm 0.0$ |
| Orig | $0.00 \pm 0.0$ | $0.60 \pm 0.2$ | $05.98 \pm 1.1$ | $10.32 \pm 1.0$ | $12.40 \pm 0.7$ | $12.75 \pm 0.5$ |
| LIRPG | $0.00 \pm 0.0$ | $0.29 \pm 0.1$ | $01.60 \pm 0.6$ | $06.10 \pm 1.3$ | $09.84 \pm 1.3$ | $10.66 \pm 1.2$ |

-----
We hope that these additional experimental results can address your concerns. Thank you again for the review!

---

### Author Response · Authors · 2022-08-09
**Response to all reviewers: Revised version of the main paper**

Dear Reviewers, Thank you again for your detailed feedback! We have now uploaded a revised version of the main paper. We believe that this revision, along with our previous responses, has addressed most of the reviewers' concerns. We truly believe that the paper has significantly improved based on the feedback, and we really appreciate the reviewers' effort. We also hope that the revised version is helpful in improving the reviewers' ratings. Below, we provide a summary of the changes:

- **Abstract:** We have clarified that the key idea of our reward shaping framework is to learn an intrinsic reward function in combination with exploration-based bonuses to maximize the agent's utility w.r.t. extrinsic rewards. We also clarified that our theoretical analysis is for a specific family of MDPs.

- **Section 2:** We have added a detailed discussion on the existing reward shaping techniques and their limitations. This discussion would provide the reader with a better perspective of the key ideas and contributions of our proposed ExpRS technique.

- **Section 3:** We have incorporated two major updates in this section. First, we have introduced Proposition 1 to emphasize our intuitive meta-gradient derivation for updating the intrinsic reward component. Second, we have updated Algorithm 2 (ExpRS) to provide a complete sketch of the overall training process, i.e., how the agent's training interleaves with reward shaping techniques.

- **Section 4:** We have incorporated two major updates in this section based on the reviewers' suggestions. First, we have added results for two new baseline techniques (Exp, LIRPG) to all the experiments as per our responses below. Second, we have now moved important environment settings from appendices to the main paper that highlight the three characteristics of (a) hard exploration, (b) local minima, and (c) noisy TV problem. In particular, the $\text{Chain}^0$ and $\text{Chain}^+$ environments in Figure 2 now correspond to $\text{Chain}$ with $n_2=40$ which was earlier only evaluated in the appendices;  the $\text{LineKey}^0$ and $\text{LineKey}^+$ environments in Figure 4c, 4d now correspond to $\text{LineKey}$ with $k=10$ keys which was earlier only evaluated in the appendices. The experimental results in Figure 2 and Figure 4 now more clearly demonstrate the effectiveness and robustness of ExpRS across different environments in comparison to all the baselines (Orig, SORS, LIRPG, Exp, SelfRS).

- **Section 5:** We have updated this section to discuss the limitations of our work and outline a future plan to address them.

Please let us know if you have any other comments or feedback. We will be happy to incorporate further feedback in the final revision. We are looking forward to hearing back from you. Thank you!

---

### Meta-Review · Area_Chair_sHap · 2022-08-27

**Recommendation:** Accept
**Confidence:** Certain

**Metareview:**

This paper proposes a self-supervised reward shaping framework. This is an interesting contribution, that is well validated theoretically and empirically. The paper is generally well-written.

I commend the authors for their helpful and positive interaction during the rebuttal period. I believe that this sufficiently addressed most of the concerns of the reviewers, particularly after the updated version. The only remaining concern for me is the lack of experiments on larger domains more commonly used in the field, but I do not consider this a critical flaw of the work. On the other hand, this paper introduces novel ideas which are well formulated and supported.

My view is that this would be a positive contribution to NeurIPS.

**Award:**

No

---

### Decision · Program_Chairs · 2022-09-14

Accept